# Hiding in Plain Sight: Disguising Data Stealing Attacks in Federated Learning

**Kostadin Garov**[1]     **Dimitar I. Dimitrov**[1,2]     **Nikola Jovanović**[2]     **Martin Vechev**[2]

[1] INSAIT, Sofia University "St. Kliment Ohridski"     [2] ETH Zurich

{kostadin.garov, dimitar.iliev.dimitrov}@insait.ai [1]

{nikola.jovanovic, martin.vechev}@inf.ethz.ch [2]

## Abstract

Malicious server (MS) attacks have enabled the scaling of data stealing in federated learning to large batch sizes and secure aggregation, settings previously considered private. However, many concerns regarding the client-side detectability of MS attacks were raised, questioning their practicality. In this work, for the first time, we thoroughly study client-side detectability. We first demonstrate that all prior MS attacks are detectable by principled checks, and formulate a necessary set of requirements that a practical MS attack must satisfy. Next, we propose SEER, a novel attack framework that satisfies these requirements. The key insight of SEER is the use of a secret decoder, jointly trained with the shared model. We show that SEER can steal user data from gradients of realistic networks, even for large batch sizes of up to 512 and under secure aggregation. Our work is a promising step towards assessing the true vulnerability of federated learning in real-world settings.

## 1 Introduction

Federated learning (FL, McMahan et al. [2017]) was proposed as a way to train machine learning models while preserving client data privacy. Recently, FL has seen a dramatic increase in real-world deployment [McMahan & Ramage; Paulik et al., 2021; FedAI]. In FL, a *server* trains a *shared model* by applying aggregated gradient updates, received from numerous *clients*.

**Gradient leakage attacks**    A long line of work [Zhu et al., 2019; Geiping et al., 2020; Zhu & Blaschko, 2021; Geng et al., 2021; Yin et al., 2021] has shown that even passive servers can reconstruct client data from gradients, breaking the key privacy promise of FL. However, these attacks are only applicable to naive FL deployments [Huang et al., 2021]—in real-life settings with no unrealistic assumptions, they are limited to small batch sizes with no secure aggregation [Bonawitz et al., 2016]. In response, recent work has argued that the honest-but-curious threat model underestimates the risks of FL, as real-world servers can be malicious or compromised. This has led to *malicious server (MS)* attacks, which have demonstrated promising results by lifting honest attacks to large batch sizes.

Most prior MS attacks rely on one of two key underlying principles. One attack class [Boenisch et al., 2021; Fowl et al., 2022b; Zhao et al., 2023; Zhang et al., 2023] uses malicious model modifications to encourage sparsity in dense layer gradients, enabling the application of analytical honest attacks—we refer to these as *boosted analytical* attacks. Other attacks utilize *example disaggregation* [Pasquini et al., 2022; Wen et al., 2022], reducing the effective batch size in the gradient space by restricting gradient flow, which permits the use of optimization-based honest attacks.

**Client-side detectability**    Nearly all prior work in the field [Geiping et al., 2020; Boenisch et al., 2021; Fowl et al., 2022b; Pasquini et al., 2022; Wen et al., 2022; Fowl et al., 2022a; Chu et al., 2023; Zhao et al., 2023] raised the issue of *client-side detectability* of MS attacks, i.e., an FL client may be able to detect malicious server activity, and decide to opt out of the current or future rounds. Despite such concerns, no attempts were made to study, quantify, or reduce the detectability of MS attacks.

**This work: detecting and disguising malicious server attacks**    We thoroughly study the question of client-side detectability of MS attacks. We demonstrate that while boosted analytical and example disaggregation attacks pose a real threat as zero-day exploits, now that their key principles are known, *all* current (and future) attacks from these two classes are client-side detectable in a principled manner, bringing into question their practicality. Notably, we demonstrate the detectability of (the more promising) example disaggregation attacks by introducing D-SNR, a novel vulnerability metric.

We observe that such limitations of prior MS attacks arise from their fundamental reliance on the honest attacks they lift. Namely, boosted analytical attacks always require handcrafted modifications which are *weight space detectable*, and example disaggregation attacks rely on the success of disaggregation, which is equally evident to any party observing the gradients, i.e., it is *gradient space detectable*. This illustrates the need for fundamentally different attack approaches.

As a step in that direction, we propose a novel attack framework SEER, which recovers data from batch sizes up to 512, yet is by design harder to detect than prior attacks. Our key insights are that (i) gradient space detection can be evaded using a *secret decoder*, disaggregating the data in a space unknown to clients, and (ii) jointly optimizing the decoder and the shared model with SGD on auxiliary data *avoids handcrafted modifications* and allows for effective reconstruction. Importantly, SEER does not lift any prior honest attack and does not require restrictive assumptions such as architecture tweaking, side-channel information, or knowledge of batch normalization data or labels.

**Key contributions**    Our main contributions are:

- We demonstrate that both boosted analytical and example disaggregation MS attacks are detectable using principled checks—for the latter, we introduce D-SNR, a novel gradient space metric of data vulnerability that can protect clients from unintended leakage. We formulate a necessary set of requirements for realistic MS attacks and make the case that detection should become a key concern when designing future attacks (Sec. 3).

- We propose SEER, a novel attack framework which satisfies all requirements based on malicious training of the shared model with a secret server-side decoder. SEER is harder to detect by design as it does not rely on honest attacks, avoiding previous pitfalls (Sec. 4). We provide an implementation of SEER at https://github.com/insait-institute/SEER.

- We present an extensive experimental evaluation of SEER on several datasets and realistic network architectures, demonstrating that it is able to recover private client data from batches as large as 512, even under the presence of secure aggregation (Sec. 5).

## 2    RELATED WORK

In this section, we discuss prior work on gradient leakage attacks in federated learning.

**Honest server attacks**    *Optimization-based attacks* [Zhu et al., 2019; Zhao et al., 2020; Geiping et al., 2020; Geng et al., 2021; Wu et al., 2021; Yin et al., 2021] optimize a dummy batch to match the user gradient. *Analytical attacks* [Phong et al., 2018; Kariyappa et al., 2022] recover inputs of linear layers in closed form, but are limited to batch size $B = 1$ and do not support convolutional networks. *Recursive attacks* [Zhu & Blaschko, 2021] extend analytical attacks to convolutional networks but are limited to $B \leq 5$. Several works thoroughly study all three attack classes [Yue et al., 2022; Balunovic et al., 2022b; Jin et al., 2021; Huang et al., 2021]. Crucially, Huang et al. [2021] show that in realistic settings, where clients do not provide batchnorm statistics and labels, all honest attacks are limited to $B < 32$ for low-res data, and fail even for $B = 1$ on high-res data. This implies that large $B$ and secure aggregation Bonawitz et al. [2016] are effective protections against honest attacks.

**Malicious server (MS) attacks**    We focus on broadly applicable boosted analytical [Boenisch et al., 2021; Fowl et al., 2022b; Zhao et al., 2023; Zhang et al., 2023] and example disaggregation attacks [Wen et al., 2022; Pasquini et al., 2022], discussed in Sec. 3. Here, we reflect on other MS attacks that study more specific or orthogonal settings. Several studies [Pasquini et al., 2022; Zhao et al., 2023] require the ability to send a different update to each user, which was shown easy to mitigate with reverse aggregation [Pasquini et al., 2022]. Lam et al. [2021] focuses on the rare setting with participation side-channel data. While we target image reconstruction, some works consider other modalities, such as text [Balunovic et al., 2022a; Gupta et al., 2022; Fowl et al., 2022a; Chu et al., 2023] or tabular data [Wu et al., 2022; Vero et al., 2022]. Further, while we focus on the threat of data reconstruction, Pasquini et al. [2022] studies weaker privacy notions such as membership [Ye et al., 2022] or property inference [Melis et al., 2019]. Finally, sybil-based attacks are a notably stronger threat model orthogonal to our work [Fung et al., 2020; Boenisch et al., 2023]. We further detail our exact threat model in App. B.

## 3    DETECTABILITY OF EXISTING MALICIOUS SERVER ATTACKS

Most malicious server (MS) attacks rely on one of two strategies based on which we group them into two classes—*boosted analytical* and *example disaggregation*. We now discuss client-side detectability of these classes and show that both are detectable with principled checks. We identify the root cause of detectability and formulate necessary requirements that future attacks must satisfy to be practical.

**Boosted analytical attacks**    The works of Boenisch et al. [2021], Fowl et al. [2022b], Zhao et al. [2023], and Zhang et al. [2023] use model modifications to induce different variants of sparsity in dense layer gradients, enabling the application of honest analytical attacks to batch sizes beyond one. Applying such attacks to the realistic case of convolutional networks requires highly unusual *architectural modifications*, i.e., placing a large dense layer in front, which makes the attack obvious. The only alternative way to apply these attacks is to set all convolutions to identity, such that the inputs are transmitted unchanged to the dense layer. As this is a pathological case that never occurs naturally and requires handcrafted changes to almost all parameters (e.g., 98% of weights in ResNet18), this approach is easily detectable by inspecting model weights (e.g., by searching for convolutional filters with a single nonzero entry, see App. A). More importantly, high levels of transmission are, in fact, impossible in realistic networks due to pooling and strides [Fowl et al., 2022b], and further attempts to conceal the changes (e.g., by adding weight noise) would additionally worsen the results.

**Example disaggregation attacks**    While the detectability of boosted analytical attacks was recognized in prior work [Geiping et al., 2020; Boenisch et al., 2021; Wen et al., 2022], example disaggregation attacks [Wen et al., 2022; Pasquini et al., 2022] are considered more promising. These attacks use model modifications to restrict the gradient flow for all but one example, causing the aggregated gradient of a batch to be equal to the gradient of a single example. This undoes the protection of aggregation and allows the attacker to apply honest optimization-based attacks to reconstruct that example. While most instantiations of example disaggregation attacks rely on unusual handcrafted parameter changes, which are detectable in the *weight space* (as for boosted analytical attacks), it may be possible to design variants that better disguise the gradient flow restriction. For this reason, we focus on a more fundamental limitation of all (current and future) example disaggregation attacks and demonstrate it makes them easily detectable in the *gradient space*. Moreover, such detection is possible without running costly optimization-based attacks by using a simple principled metric.

We now propose one such metric, the *disaggregation signal-to-noise ratio (D-SNR)*. Assuming the use of the standard cross-entropy loss $\mathcal{L}(x, y)$, a shared model with parameters $\boldsymbol{\theta}$, and a batch of data $D = \{(x_1, y_1), \ldots, (x_B, y_B)\}$, we define D-SNR as follows:

$$D\text{-}SNR(\boldsymbol{\theta}, D) = \max_{W \in \boldsymbol{\theta}_{lw}} \frac{\max_{i \in \{1,\ldots,b\}} \left\| \frac{\partial \mathcal{L}(x_i, y_i)}{\partial W} \right\|}{\sum_{i=1}^{b} \left\| \frac{\partial \mathcal{L}(x_i, y_i)}{\partial W} \right\| - \max_{i \in \{1,\ldots,b\}} \left\| \frac{\partial \mathcal{L}(x_i, y_i)}{\partial W} \right\|} \tag{1}$$

where $\boldsymbol{\theta}_{lw}$ denotes the set of weights of all linear layers (dense and convolutional; 98% of ResNet18). Intuitively, D-SNR searches for layers where the batch gradient (the average of example gradients) is dominated by the gradient of a single example, suggesting disaggregation. We conservatively use $\max$ to avoid false negatives and account for attempts at partial disaggregation, i.e., if there is *any* layer that disaggregates a single example, D-SNR will be large, and the client may decide that their batch is vulnerable and skip the current training round. While we focus on the case of disaggregating a single example, our approach can be easily generalized to any number of examples.

We use D-SNR to experimentally study the detectability of example disaggregation attacks in realistic settings (see App. A for experimental details). As D-SNR is always $\infty$ for attacks proposed by Wen et al. [2022], we modify them in an attempt to smoothly control the strength of the gradient flow restriction. Our key observation, presented in Fig. 1 (red ✗), is that in all cases where the attack is successful, D-SNR is unusually large, making the attack easily detectable. Reducing the strength of the gradient flow restriction further causes a sharp drop in D-SNR, entering the range of most non-malicious networks (blue ✗), i.e., the attack is undetectable. However, in all such cases, the attack fails, as the aggregation protects the examples. In rare cases (e.g., when overfitting), even natural networks can produce high D-SNR and be flagged. This behavior *is desirable*, as such networks indeed disaggregate a single example, and (unintentionally) expose sensitive user data. Thus, metrics such as D-SNR should be interpreted as detecting *vulnerability*, and not necessarily *maliciousness*.

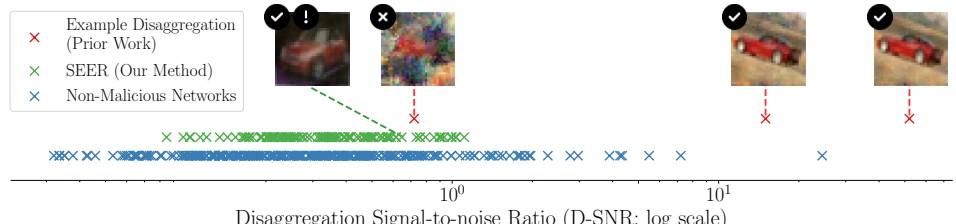

Figure 1: D-SNR (Sec. 3) of real (model, data batch) pairs. High values indicate vulnerability to data leakage, which can manifest even in non-malicious models (✕). Example disaggregation attacks (✗) are easily detectable as they can successfully reconstruct data (✅) only when DSNR is unusually high (note the log scale), and fail otherwise (❌). Our method, SEER (✗, Sec. 4), successfully reconstructs an example even when D-SNR is low (✅ ❗), and is thus hard to detect in the original gradient space.

**Requirements for future attacks**   Our results show that all prior MS attacks are client-side detectable using generic checks. We argue that this is caused by fundamental problems in the design principles of the two attack classes and can not be remedied by further refinements. Any attempt to lift an honest analytical attack will inherit the limitation of being inapplicable to convolutions and will require architectural changes or handcrafted modifications detectable in the weight space. Lifting optimization-based attacks always requires example disaggregation, which is gradient space detectable. More broadly, as all information needed to execute these attacks is in the user gradients, the server has no informational advantage and no principled way to conceal the malicious intent.

This suggests that new attack principles are required to better exploit the potential of the MS threat model. To help guide the search, we now state the necessary requirements for future MS attacks guided by our results above and observations from prior work [Wen et al., 2022; Huang et al., 2021]. We argue that realistic MS data stealing attacks for image classification should: (i) target realistic deep convolutional networks with large batch sizes and/or secure aggregation; (ii) only utilize the attack vector of weight modifications, with no protocol changes (e.g., non-standard architectures, asymmetric client treatment) and no sybil capabilities; (iii) not assume unrealistic side information, such as batch normalization statistics or label information [Huang et al., 2021]; and (iv) explicitly consider the aspects of weight and gradient space detection (e.g., avoid obvious handcrafted modifications).

## 4    SEER: DATA STEALING VIA SECRET EMBEDDING AND RECONSTRUCTION

In this section, we propose SEER, a novel attack framework that steals data from large batches while satisfying the requirements in Sec. 3. SEER avoids both pitfalls of prior MS attacks that caused them to be detectable. Namely, SEER does not lift any honest attack and evades gradient space detection by disaggregating the data in a *hidden space* defined by a server-side *secret decoder*. As a result, SEER-trained networks (green ✗) have D-SNR values indistinguishable from those of natural networks (Fig. 1). Further, SEER does not use handcrafted modifications, and instead trains the shared model and the secret decoder jointly with SGD, evading weight space detection.

**Overview**   Once trained, SEER is mounted as follows (Fig. 2). As in standard FL, the client propagates their batch $(\boldsymbol{X}, \boldsymbol{y})$ of $B$ examples $(\boldsymbol{x}_i, \boldsymbol{y}_i)$ through the shared model $f$ with parameters $\boldsymbol{\theta}_f$ sent by the server, and returns the gradient $\boldsymbol{g}$ (for simplicity we assume FedSGD) of the public loss $\ell$ w.r.t. $\boldsymbol{\theta}_f$. When $B > 1$, $\boldsymbol{g}$ aggregates gradients $\boldsymbol{g}_i$ of individual examples, i.e., $\boldsymbol{g} = (1/B) \sum_{i=1}^{B} \boldsymbol{g}_i$. When *secure aggregation* is used (discussed shortly), the sum also includes gradients of other clients.

The server's goal is to break this aggregation. To this end, the server feeds $\boldsymbol{g}$ to a *secret decoder* consisting of a *disaggregator* $d$, followed by a *reconstructor* $r$. Crucially, $d$ is trained to project $\boldsymbol{g}$ onto a hidden space in which the gradient projections of all images not satisfying some property $\mathcal{P}$ are removed. While the exact choice of $\mathcal{P}$ is not essential, the goal is that for most batches *only one batch example satisfies* $\mathcal{P}$. In Fig. 2, $\mathcal{P} =$"images with brightness at most $\tau$" with $\tau$ chosen so only $\boldsymbol{x}_5$ satisfies it, allowing $d$ to extract the projected gradient $d(\boldsymbol{g}_5)$, and $r$ to steal the client image $\boldsymbol{x}_5$.

To train SEER, the server chooses $\mathcal{P}$ and interprets $\boldsymbol{\theta}_f, \boldsymbol{\theta}_d$, and $\boldsymbol{\theta}_r$ as an encoder-decoder framework, trained end-to-end using auxiliary data to simulate a real client. The goal of training is for $d$ to nullify the contributions of images not satisfying $\mathcal{P}$ ($\mathcal{L}_{\text{nul}}$ in Fig. 2), and for $r$ to reconstruct the image satisfying $\mathcal{P}$ from the output of $d$ ($\mathcal{L}_{\text{rec}}$). The shared model $f$ is also trained to encode client data in the gradient space in a way that supports the goals of disaggregation and reconstruction.

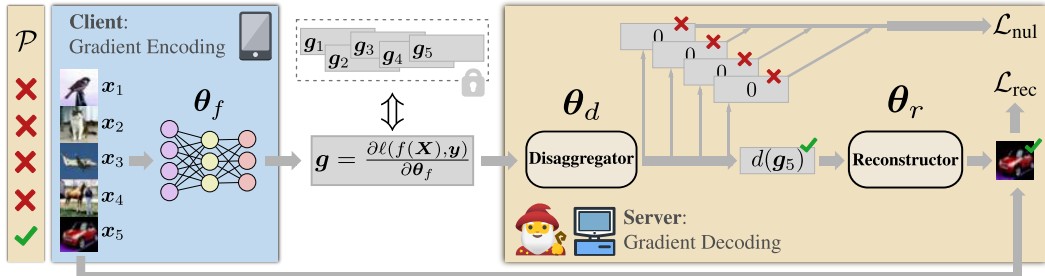

Figure 2: Overview of SEER. A client propagates a batch $X$ (of which one image satisfies the property $\mathcal{P}$ only known to the server) through the shared network $f$ with malicious weights $\theta_f$, and returns the aggregated gradient $g$, hoping that the aggregation protects individual images. The server steals the image satisfying $\mathcal{P}$ by applying a secret disaggregator $d$ to remove the impact of other images in a hidden space, followed by a secret reconstructor $r$. SEER is trained by jointly optimizing $\theta_f$, $\theta_d$, and $\theta_r$ to minimize a weighted sum of $\mathcal{L}_{\text{nul}}$ and $\mathcal{L}_{\text{rec}}$.

## 4.1 KEY COMPONENTS OF SEER

We next describe the individual components of SEER in more detail.

**Selecting the property** $\mathcal{P}$   Let $I_{\text{nul}} \subseteq [B] \coloneqq \{1, \ldots, B\}$ denote the set of examples in the client batch that do not satisfy the secret property $\mathcal{P}$, and $I_{\text{rec}} \subseteq [B]$ the set of those that do. Following Fowl et al. [2022b], we use properties of the type $m(\boldsymbol{x}) < \tau$, where $m$ represents some image measurement, e.g., brightness, and $\tau$ is chosen to maximize $P(|I_{\text{rec}}| = 1)$. Similarly to Fowl et al. [2022b], in our experiments we use $m$ based on brightness and color, but we emphasize that the attacker can use many different types of $m$ (as we experimentally demonstrate in App. F.6), i.e., this choice is non-restrictive and is simply a way to single out images from big batches.

Fowl et al. [2022b] propose the *global* setting, where the server sets a single value of $\tau$ for all client batches. In particular, they choose $\tau$ such that the probability of satisfying $\mathcal{P}$ is $1/B$ on the distribution of $m$ over the whole space of input examples. Wen et al. [2022] later showed that this choice is optimal in the global setting, resulting in $P(|I_{\text{rec}}| = 1) \to 1/e$ from above as $B \to \infty$. We improve upon this by allowing $\tau$ to be dependent on the client batch data $(\boldsymbol{X}, \boldsymbol{y})$, i.e., we propose a *local* setting. We make the novel observation that when batch normalization (BN) is present (like in most convolutional networks), we can choose $\mathcal{P}$ with respect to the local (in-batch) distribution of $m$, e.g., as the minimal brightness in every batch. We find that SEER can be trained on such local $\mathcal{P}$ with auxiliary data, empirically achieving $P(|I_{\text{rec}}| = 1) > 0.9$ for $B$ as large as $512$, which is a significant improvement over the probability of $1/e$ achieved in the global setting. Our method is enabled by the fact that each BN layer normalizes the distribution of its input, intertwining the computational graphs of images in the batch which are otherwise independent. Unlike prior work, our local properties $\mathcal{P}$, allow the attacker to, most of the time, steal the client data after only a single communication round.

We note that *secure aggregation* [Bonawitz et al., 2016] is more challenging, and *not equivalent* to a large batch from one client when BN is present, an aspect overlooked in prior work. To overcome this, we design a more elaborate $\mathcal{P}$ that combines local and global properties, resulting in $P(|I_{\text{rec}}| = 1) \to 1/e$, as in prior work. We provide a detailed explanation in App. C, and in our experiments in Sec. 5 evaluate both large batch and secure aggregation variants of SEER.

**Training $\theta_f$ for suitable gradient encodings**   Training the shared model weights $\theta_f$ along with the secret decoder is essential for the success of SEER. Intuitively, we can interpret the client-side gradient computation as a latent space encoding of the client data. The failures of honest attacks, discussed in Sec. 2, suggest that the gradient encoding often lacks the required information to reconstruct user data. Our key observation is that the MS threat model uniquely allows to overcome this issue by *controlling the gradient encoding* by tuning $\theta_f$. In particular, we maliciously optimize $\theta_f$ with SGD to allow the recovery of a single example by the other modules of SEER, regardless of the information lost at the encoding step. While Zhang et al. [2023] also considered optimization-based modifications with auxiliary data, their approach still inherits the fundamental limitations of all boosted analytical attacks, requiring additional handcrafted modifications which, as noted in Sec. 3, are easily weight space detectable—an issue that SEER circumvents by design.

**Training $\theta_d$ for secret disaggregation**    The secret disaggregator $d$ addresses the key limitation of example disaggregation attacks (discussed in Sec. 3), i.e., disaggregating examples in the gradient space. In contrast, $d$ embeds the gradients from $g$ into a lower-dimensional space $\mathbb{R}^{n_d}$ using a secret linear map $\theta_d$, concealing the disaggregation in the original gradient space. The benefits of using such a linear map are twofold. First, the linear map commutes with gradient aggregation due to additivity. Second, the lower-dimensional space allows us to more easily drive the projected gradients of $I_{\text{nul}}$ to $0$, which happens when they are in or close to the null space of $\theta_d$. Combining the two properties (($i$) and ($ii$) in Eq. 2) ideally allows us to retain only the chosen sample from the aggregated gradient $g$:

$$d(g) = d(\sum_{i=1}^{B} g_i) \stackrel{(i)}{=} \sum_{i=1}^{B} d(g_i) = \sum_{i \in I_{\text{nul}}} d(g_i) + \sum_{i \in I_{\text{rec}}} d(g_i) \stackrel{(ii)}{\approx} \sum_{i \in I_{\text{rec}}} d(g_i). \tag{2}$$

To achieve this in practice, $f$ and $d$ should be set such that $d(g_i) \approx 0$ for all $i \in I_{\text{nul}}$, while tolerating $d(g_i) \neq 0$ for the single $i \in I_{\text{rec}}$. To this end, for $\mathcal{P}$ chosen as discussed above, we define the following objective:

$$\mathcal{L}_{\text{nul}} = \sum_{i \in I_{\text{nul}}} \| d(g_i) \|_2^2, \tag{3}$$

which SEER aims to minimize during training. We ensure that this does not also nullify $d(g_i)$ for the example of interest in $I_{\text{rec}}$, so $r$ is able to recover that example from $d(g_i)$, as described next.

---

**Algorithm 1** The training procedure of SEER

1: **function** TRAINSEER($f$, $\ell$, $B$, $\mathcal{X}$, $\mathcal{Y}$)
2:     Choose $\mathcal{P}$, initialize $d$ and $r$
3:     **while** not converged **do**
4:         $X, y \leftarrow \{x_i, y_i \sim (\mathcal{X}, \mathcal{Y}) \,|\, i \in [B]\}$
5:         $I_{\text{nul}}, I_{\text{rec}} \leftarrow \mathcal{P}(X, y)$
6:         $X_{\text{nul}}, y_{\text{nul}} \leftarrow X[I_{\text{nul}}], y[I_{\text{nul}}]$
7:         $X_{\text{rec}}, y_{\text{rec}} \leftarrow X[I_{\text{rec}}], y[I_{\text{rec}}]$
8:         $g_{\text{nul}}, g_{\text{rec}} \leftarrow$ BP($f, \ell, X_{\text{nul}}, X_{\text{rec}}, y_{\text{nul}}, y_{\text{rec}}$)
9:         $\mathcal{L}_{\text{nul}} \leftarrow \| d(g_{\text{nul}}) \|_2^2$    ▷ Eq. 3
10:        $\mathcal{L}_{\text{rec}} \leftarrow \| r(d(g_{\text{rec}})) - X_{\text{rec}} \|_2^2$    ▷ Eq. 4
11:        $\mathcal{L} \leftarrow \mathcal{L}_{\text{rec}} + \alpha \cdot \mathcal{L}_{\text{nul}}$    ▷ Eq. 5
12:        $\theta_m \leftarrow \theta_m - \gamma_m \cdot \frac{\partial \mathcal{L}}{\partial \theta_m}, \forall m \in \{f, d, r\}$
13:     **end while**
14:     **return** $f, d, r$
15: **function** BP($f$, $\ell$, $X_{\text{nul}}$, $X_{\text{rec}}$, $y_{\text{nul}}$, $y_{\text{rec}}$)
16:     $[\![l_{\text{nul}}; l_{\text{rec}}]\!] \leftarrow \ell(f([\![X_{\text{nul}}; X_{\text{rec}}]\!]), [\![y_{\text{nul}}; y_{\text{rec}}]\!])$
17:     **return** $\frac{\partial l_{\text{nul}}}{\partial \theta_f}, \frac{\partial l_{\text{rec}}}{\partial \theta_f}$

---

**Training $\theta_r$ for image reconstruction** The final component of SEER we discuss is the secret reconstructor $r: \mathbb{R}^{n_d} \to \mathbb{R}^{n_r}$, which receives $d(g)$, i.e., the (noisy) isolated embedding of the target image gradient, as seen in Eq. 2. The reconstructor aims to map $d(g)$ back to the original image $x_{\text{rec}}$, effectively stealing that example from the original batch, compromising client privacy. To this end, we define the following $\ell_2$ reconstruction objective, which is at odds with $\mathcal{L}_{\text{nul}}$:

$$\mathcal{L}_{\text{rec}} = \| r(d(g_{\text{rec}})) - x_{\text{rec}} \|_2^2. \tag{4}$$

The final loss function of SEER weighs the two losses using a hyperparameter $\alpha > 0$:

$$\mathcal{L} = \mathcal{L}_{\text{rec}} + \alpha \cdot \mathcal{L}_{\text{nul}}. \tag{5}$$

All three key components of SEER are jointly trained to minimize $\mathcal{L}$.

## 4.2 END-TO-END ATTACK DESCRIPTION & DISCUSSION

Algorithm 1 describes the training of SEER. We train on client-sized batches (see App. F.3 for a related study) sampled from our auxiliary data (Line 4). Based on $\mathcal{P}$, we select the index sets $I_{\text{nul}}$ and $I_{\text{rec}}$ (Line 5), representing the examples we aim to disaggregate. Then, we simulate the client updates $g_{\text{rec}}$ and $g_{\text{nul}}$ computed on the full batch $X$ (Line 8), and use them to compute our optimization objective (Line 11). We minimize the objective by jointly training $f$, $d$, and $r$ using SGD (Line 12).

Mounting SEER once the malicious weights $\theta_f$ have been trained using Algorithm 1 is simple, as we illustrate in Algorithm 2. The server, during an FL round, sends the client the malicious model $f$ (Line 2), and receives the gradient update $g$. Then, it applies its secret disaggregator $d$ and reconstructor $r$ (Line 3) to obtain $x_{\text{stolen}}$, the reconstructed private example from the client batch.

---

**Algorithm 2** Mounting SEER

1: **function** MOUNTSEER($f$, $d$, $r$)
2:     $g \leftarrow$ GETCLIENTUPDATE($f$)
3:     $x_{\text{stolen}} \leftarrow r(d(g))$
4:     **return** $x_{\text{stolen}}$

---

Table 1: Large batch reconstruction for different batch sizes $B$. The metrics are introduced at the top of Sec. 5. Results with 2 more settings (CIFAR100, Bright and CIFAR10, Red) are given in App. F.1.

| | CIFAR10, Bright | | | CIFAR100, Red | | |
|---|---|---|---|---|---|---|
| $B$ | Rec (%) | PSNR-Top ↑ | PSNR-All ↑ | Rec (%) | PSNR-Top ↑ | PSNR-All ↑ |
| 64 | 89.4 | $32.1 \pm 2.0$ | $27.2 \pm 5.3$ | 97.1 | $31.7 \pm 1.1$ | $29.0 \pm 3.4$ |
| 128 | **94.2** | $31.9 \pm 1.7$ | $28.2 \pm 4.3$ | 97.4 | $31.8 \pm 1.1$ | $29.3 \pm 3.2$ |
| 256 | 93.5 | $\mathbf{32.8 \pm 2.0}$ | $\mathbf{28.5 \pm 5.0}$ | 97.7 | $31.3 \pm 1.0$ | $28.6 \pm 3.2$ |
| 512 | 87.8 | $26.6 \pm 1.8$ | $23.2 \pm 3.5$ | **98.6** | $\mathbf{33.1 \pm 1.1}$ | $\mathbf{30.5 \pm 3.1}$ |

**SEER satisfies all requirements**  We now reflect on the requirements listed in Sec. 3 and discuss how SEER satisfies them. First, SEER does not utilize any attack vector apart from maliciously modifying the weights of $f$, does not assume unrealistic knowledge of BN statistics or batch labels, and makes no assumptions regarding label distributions, in contrast with some prior work [Yin et al., 2021; Wen et al., 2022]. We remark that the necessity of such side information is the artifact of optimization-based attacks, and another reason why approaches that do not attempt to lift honest attacks (such as SEER) may be more promising. SEER was greatly influenced by the assumption that clients *will* inspect the models, aiming to detect malicious updates. Namely, SEER avoids weight space detectable handcrafted modifications and introduces secret disaggregation as means to also avoid gradient space detection. As we show in Sec. 5, SEER successfully steals client data on realistic convolutional networks with large batch sizes and secure aggregation, demonstrating its practicality.

## 5   EXPERIMENTAL EVALUATION

In this section, we present our experimental results, demonstrating that SEER is effective at reconstructing client images from realistic networks, in both large batch and secure aggregation settings. These results are especially valuable given the important advantages of SEER over prior work in terms of satisfying the requirements for practical attacks (Sec. 3), as we have discussed in Sec. 4.

**Experimental setup**  We use ResNet18 [He et al., 2016] in all experiments. We use the *CIFAR10* dataset, as well as *CIFAR100* [Krizhevsky et al., 2009] and *ImageNet* [Deng et al., 2009], to demonstrate the ability of SEER to scale with the number of labels and input size, respectively. We generally use the training set as auxiliary data, and mount the attack on randomly sampled batches of size $B$ from the test set for CIFAR10/100 and validation set for ImageNet. We further experiment with auxiliary datasets of different sizes in App. F.8, and clients with different heterogeneity levels in App. F.9 where we show that SEER is highly effective even when only small amount of auxiliary data is available and when clients data is highly heterogeneous. We run all experiments on a single NVIDIA A100 GPU with $40$GB (CIFAR10/100) and $80$GB (ImageNet) of VRAM. Each CIFAR experiment took $< 7$ GPU days to train and $< 1$h to mount on 1000 batches. The ImageNet model trained for 14 GPU days, with $0.5h$ to mount the attack on 100 batches. In our CIFAR experiments, we set $r$ to a linear layer and subsume $d$ in it. For ImageNet, we use a linearized U-Net decoder [Ronneberger et al., 2015] (see App. G). We defer additional implementation details to App. H and App. I.

In all experiments, we use the properties of maximal brightness (*Bright*) and redness (*Red*), training separate malicious weights for each (dataset, property, batch size) triple. We report 3 reconstruction quality metrics: (i) the fraction of good reconstructions (*Rec*), i.e., batches where reconstructions have PSNR $> 19$ [Horé & Ziou, 2010] to the ground truth; (ii) the average PSNR across all attacked batches (*PSNR-All*); and (iii) the average PSNR for the top $\frac{1}{e} \approx 37\%$ of the batch reconstructions (*PSNR-Top*) that allows to compare SEER in large batch (1 client) and secure aggregation (many clients) settings. We provide experiments with more properties and metrics in App. F.6 and App. J.

**Large batch reconstruction on CIFAR10/100**  A subset of our main results is shown in Table 1; the full results are deferred to App. F.1 and follow similar trends. We make several key observations. First, in most experiments, the use of local properties (see Sec. 4.1) allows us to steal an image from most batches (up to 98.6%), greatly improving over $1/e\%$ achieved by prior work. Second, we obtain good reconstructions for both Red and Bright property (average PSNR up to 30), which confirms that SEER can handle a diverse set of properties, and that property choice is not crucial for its success. Finally, SEER successfully steals images even from very large batch sizes such as 512, showing no clear degradation in performance. On top of these quantitative results, we show example reconstructions in Fig. 3 (left, $C = 1$), visually confirming their quality.

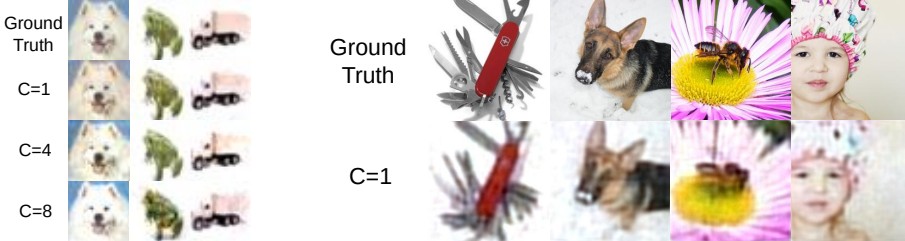

Figure 3: Example reconstructions of SEER with 128 total examples and different number of clients $C$ on CIFAR10 (Left) and 64 examples on ImageNet (Right), both using the Bright property.

**Large batch reconstruction on ImageNet**  To show scalability to high-res images, we train SEER on ImageNet with $B = 64$ and the Bright property (due to high computational costs, we leave more thorough studies of ImageNet to future work). To speed up convergence, we pretrain parts of the transposed convolution stack alongside $\theta_f$ on downsized images, and use it as initialization. We obtain PSNR-All of $21.9 \pm 3.5$ and PSNR-Top of $25.1 \pm 2.5$, corresponding to $82.1\%$ successfully attacked batches. Our results are significant, as these PSNR values on Imagenet represent very high quality reconstructions and are higher than the state-of-the-art attack in Wen et al. [2022]. This is further visually confirmed by the example reconstructions we show in Fig. 3 (right). From the recovered images we conclude that SEER can be efficiently instantiated on high-resolution images, resulting in very detailed reconstructions that allow the identification of complex objects and individual people, constituting a serious violation of privacy. We note the significance of these results, as stealing even a single ImageNet image is impossible with honest attacks without restrictive assumptions [Huang et al., 2021]. We see these results as an encouraging signal for general applicability of SEER.

**Secure aggregation**  In Table 2, we present the results of CIFAR10 experiments with secure aggregation with $C = 4$ and $C = 8$ clients and batch sizes chosen to match the total number of images (*#Imgs*) as in Table 1. Most importantly, as before, SEER consistently obtains image reconstructions with average PSNR $> 25$, i.e., recovers most images almost perfectly. Comparing to Table 1, the success probability degrades with $C$, confirming our intuition (see Sec. 4) that secure aggregation

Table 2: CIFAR10 reconstruction with secure aggregation, varying the number of clients ($C$) and total images (#Imgs). See App. F.2 for results with another property.

| #Imgs | $C = 4$, Bright | | $C = 8$, Bright | |
|---|---|---|---|---|
| | Rec (%) | PSNR-Top $\uparrow$ | Rec (%) | PSNR-Top $\uparrow$ |
| 64 | 41.4 | **27.3 ± 3.1** | 41.3 | 26.6 ± 3.7 |
| 128 | 44.2 | 26.8 ± 3.0 | 40.6 | **27.3 ± 3.3** |
| 256 | 51.9 | **27.3 ± 2.5** | 41.9 | 25.4 ± 3.1 |
| 512 | **52.9** | 25.7 ± 2.4 | **51.7** | 25.9 ± 2.8 |

provides additional protection in the presence of BN, compared to simply using large batches. Despite this, the success probability Rec is significantly higher than $1/e\%$ of prior work. We suspect this is due to the model learning a restricted version of single-client reconstruction for each client, and further compare the two variants in App. F.5. We note that Rec rises with the number of images, which we believe is due to the better estimation of the property threshold for larger batches. Finally, in Fig. 3 (left), we can visually compare results for different number of clients, noting no obvious degradation, which reaffirms that SEER can breach privacy even when secure aggregation is used.

**Robustness to distribution shifts**  A question that naturally arises is if the need for auxiliary data restricts the applicability of SEER. In this experiment we show otherwise, demonstrating robustness to distribution shifts between the attacker's auxiliary dataset ($D_a$) and the client dataset ($D_c$), i.e., an attacker can successfully mount SEER without the knowledge of $D_c$, relying only on public data. We set $C = 1, B = 128$, and $D_a =$ CIFAR10 and explore several options for $D_c$, illustrating dif-

Table 3: SEER is robust to distribution shifts between the auxiliary dataset (CIFAR10) and the client dataset $D_c$. We use $B = 128$ and the Red property.

| $D_c$ | Rec (%) | PSNR-Top $\uparrow$ | PSNR-All $\uparrow$ |
|---|---|---|---|
| CIFAR10 | 93.5 | 31.1 ± 1.2 | 27.8 ± 4.1 |
| CIFAR10.1v6 | **96.0** | **31.6 ± 1.0** | **28.4 ± 3.8** |
| CIFAR10.2 | 90.2 | 31.6 ± 1.3 | 27.5 ± 5.0 |
| TinyImageNet | 80.2 | 27.6 ± 1.0 | 23.7 ± 4.7 |
| ISIC2019 | 98.0 | 29.4 ± 1.0 | 26.9 ± 2.84 |

ferent levels of shift. Namely, CIFAR10.1v6 [Recht et al., 2018] and CIFAR10.2 [Lu et al., 2020] represent naturally occurring shifts of the data source, TinyImageNet [Le & Yang, 2015] (mapped to 10 classes) models different data sources for $D_a$ and $D_c$, and ISIC2019 [Tschandl et al., 2018; Codella et al., 2018; Combalia et al., 2019] models a more severe domain shift between $D_a$ and $D_c$.

Table 4: Comparison between SEER and prior state-of-the-art MS attacks.

| Method | Und-Rec (%) | PSNR-Und-Rec ↑ | PSNR-Und ↑ | Rec (%) | PSNR-All ↑ |
|---|---|---|---|---|---|
| Fishing $\beta = 400$ | 4 | $20.2 \pm 0.5$ | $17.1 \pm 2.3$ | 77 | $21.7 \pm 3.2$ |
| Fishing $\beta = 100$ | 8 | $20.6 \pm 1.6$ | $16.4 \pm 2.9$ | 63 | $20.2 \pm 4.1$ |
| Fishing $\beta = 50$ | 4 | $19.4 \pm 0.3$ | $15.5 \pm 2.2$ | 52 | $19.4 \pm 4.5$ |
| Fishing $\beta = 12.5$ | 1 | $22.4 \pm 0.0$ | $13.7 \pm 2.0$ | 8 | $14.5 \pm 3.4$ |
| Zhang23 | 0 | $N/A$ | $N/A$ | 5 | $15.8 \pm 1.8$ |
| LOKI | 0 | $N/A$ | $N/A$ | 100 | $\mathbf{143.4 \pm 10.3}$ |
| SEER | **90** | $\mathbf{24.6 \pm 2.2}$ | $\mathbf{23.8 \pm 3.3}$ | 90 | $23.8 \pm 3.3$ |

The results are shown in Table 3. We observe no degradation for CIFAR10.1v6 and CIFAR10.2, confirming that SEER can handle naturally-occurring data shifts. For TinyImageNet and ISIC2019, despite the large discrepancy to CIFAR10 in image and label distributions, we observe high quality reconstruction on 80% and 98% of images, confirming that SEER is not limited by the choice of $D_a$. We further investigate the robustness to batch size mismatch in App. F.3 and corruptions in App. F.4.

**Comparison to prior MS attacks** We compare SEER to 3 state-of-the-art MS attacks: (i) *Fishing* [Wen et al., 2022], an example disaggregation attack; (ii) *Zhang23* [Zhang et al., 2023], a boosted analytical attack; and (iii) *LOKI* [Zhao et al., 2023], a boosted analytical attack that relies on a stronger threat model that permits architectural changes and sending different models to clients. We attack 100 batches on CIFAR10, with $C = 1, B = 128$. We use the Red property for SEER. As in Sec. 3, we explore different variants of Fishing by varying the parameter $\beta$ which should control the strength-detectability tradeoff. We report the usual metrics *Rec* and *PSNR-All*, the percentage of undetected successful attacks (*Und-Rec*) based on D-SNR (Sec. 3) and T-SNR (App. A), as well as the average PSNR of all undetected reconstructions (*PSNR-Und*), and the successful undetected reconstructions (*PSNR-Und-Rec*). We provide more details about the experimental setup in App. I.5.

The results are shown in Table 4. Setting detectability aside, SEER outperforms all methods but LOKI, while also being very fast to mount (<2 sec per batch). We emphasize that LOKI's performance is largely due to its architectural changes to the ResNet which are trivially detectable by clients and crucial to the application of the method. We also observe that Zhang23 fails to recover most images at all due to the stride>1 and pooling in realistic networks that cause severe downscaling of the image fed to the attacked linear layer as discussed in Sec. 3. Crucially, only a tiny fraction of successful Fishing attacks are undetected, while other prior methods completely fail to avoid detection. In contrast, for SEER *all* successful attacks remain undetected. Finally, we confirm our observation from Fig. 1 that prior MS attacks need to jeopardize reconstruction quality to avoid detection, as for Fishing PSNR-Und is well below PSNR-All for all values of $\beta$. Our experiments reaffirm that the reliance on honest attacks of prior MS attacks makes them easily detectable, and thus unrealistic.

## 6 OUTLOOK

While SEER is a powerful attack that can harm user privacy, we believe our work opens the door to a more principled investigation of defenses, as it illustrates that techniques such as secure aggregation are not as effective as previously thought. To mitigate attacks like ours, prior work has discussed cryptographic techniques like SMPC or FHE, which are still largely impractical [Kairouz et al., 2019], as well as differential privacy methods, which we demonstrate in App. E to not be effective enough.

Thus, we believe that principled client-side detection is the most promising way forward. While SEER avoids pitfalls of prior attacks which make them easily detectable, and we see no clear ways to detect it currently, more mature detection techniques may be able to do so. We encourage such work and advocate for efficient and robust checks based on attack categorization (such as in this work), as opposed to ad-hoc detection which attacks can easily adapt to. On the attack side, interesting future directions include other data modalities and model architectures and improving SEER's training cost.

## 7 CONCLUSION

In this work, we explored the issue of client-side detectability of malicious server (MS) attacks in federated learning. We demonstrated that all prior attacks are detectable in a principled way, and proposed SEER, a novel attack strategy that by design avoids such detection while effectively stealing data despite aggregation. Our work highlights the importance of studying attack detectability and represents a promising first step towards MS attacks that compromise privacy in realistic settings.

ETHICS STATEMENT

As we noted in Sec. 6, the attack introduced by this work, SEER, advances the capabilities of attackers aiming to compromise client privacy in FL. Further, as Wen et al. [2022] point out, attacks like ours based on property thresholding can lead to disparate impact, affecting outlier groups more severely as their inputs are more likely to be reconstructed. However, we believe that our principled investigation of detection and the emphasis on realistic scenarios, as well as making the details of our attack public and open source (which we intend to do after publication), both have a significant positive impact, as they open the door to further systematic studies of defenses, and help practitioners better estimate the privacy risks of their FL deployments and avoid the common error of underestimating the vulnerability.

ACKNOWLEDGMENTS

This research was partially funded by the Ministry of Education and Science of Bulgaria (support for INSAIT, part of the Bulgarian National Roadmap for Research Infrastructure).

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

## A    DETECTABILITY EXPERIMENTS

Here we provide more details regarding our detectability experiments.

**Measuring D-SNR**    To produce Fig. 1, we consider 4 SEER-trained malicious models (CIFAR10, *Bright/Dark*, batch size 128/256), as well as 8 checkpoints made at various points during natural training, using the same initialization as used for SEER. Then, for each value of $B \in \{16, 32, 64\}$, we choose 5 random batches of size $B$ from the training set and 5 random batches of size $B$ from the test set of CIFAR10. For each batch, we compute the D-SNR on each of the 12 networks using Eq. 1 and plot the resulting value as a point in Fig. 1 (blue for natural and green for SEER networks). For example disaggregation attacks, we use a publicly available implementation of the attacks of Wen et al. [2022] and modify the *multiplier* parameter to control the strength of the attack. We use the default setting where batches are chosen such that all images belong to the same class (*car* in this case). The three reconstructions of the example disaggregation attack are obtained by running the modernized variant of the attack of Geiping et al. [2020] on the disaggregated batch. The modernized variant is implemented in the Breaching framework, which Wen et al. [2022] is a part of. Finally, for the reconstruction of SEER, we aimed to show an image from the same class (a car), with D-SNR slightly below the D-SNR of the leftmost example disaggregation point (0.72). To do this, we use the *Dark* property and the dark car image from Fig. 3, and sample the other 63 examples in the batch randomly from the test set until the D-SNR falls in the $[0.62, 0.72]$ range. We stop as soon as we find such a batch and report the reconstruction produced by SEER.

**Measuring transmission**    As noted in Sec. 3, to be applicable to convolutional networks, boosted analytical attacks require handcrafted changes to convolutional layers that simply transmit the inputs unchanged. While, as noted above, even in the ideal case, this cannot lead to good reconstructions, we illustrate the point that such change is detectable by defining a metric similar to D-SNR, which can be interpreted as a *transmission signal-to-noise ratio*, measured on the first convolutional layer. Namely, for each filter in the first convolutional layer, we divide the absolute values of the largest entry by the sum of the absolute values of all other entries. Intuitively, we treat the entry with the largest absolute value as the signal, and measure how well this is transmitted by the filter. The ratio is high when the filter transmits the input unchanged, and is $\infty$ for the handcrafted changes used by the boosted analytical attacks. We compute this metric on the 12 networks used in Fig. 1 (see previous paragraph) and show the results in Fig. 4. Intuitively, the red line at $1.0$ indicates the case where there are equal amounts of the pixel being transmitted and all other pixels. We can observe that all networks have values below $0.3$, confirming that transmission is indeed unusual and not a case that ever happens naturally, implying that if boosted analytical attacks that use this technique would be able to obtain good results, they would still be easily detectable in the weight space.

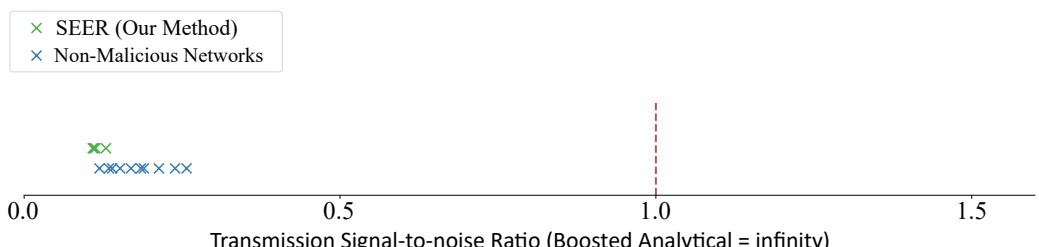

Figure 4: The transmission signal-to-noise ratio of several SEER-trained and naturally trained networks. The same metric has a value of $\infty$ for all boosted analytical attacks.

## B    PRECISE THREAT MODEL

SEER is an MS gradient leakage attack, and thus assumes the ability of the attacker to choose the weights of the model such that client data is easier to recover. Similarly to most other MS attacks [Boenisch et al., 2021; Wen et al., 2022; Zhang et al., 2023], we focus on attacking the FedSGD [McMahan et al., 2017] protocol at a single communication round, and similarly to Wen et al. [2022], we recover a single user image from a large batch of client images. Importantly,

unlike previous MS attacks, SEER does not lift any prior honest attack [Boenisch et al., 2021; Fowl et al., 2022b; Zhao et al., 2023; Zhang et al., 2023], does not require restrictive assumptions such as architecture tweaking [Zhao et al., 2023], side-channel information, or knowledge of batch normalization data or labels [Wen et al., 2022], and does not require the ability to send different updates to different users [Pasquini et al., 2022; Zhao et al., 2023]. Similarly to Fowl et al. [2022b], SEER requires auxiliary data to be available to the server. We believe this does not limit SEER's practical applicability, as we demonstrate that we can train SEER's malicious weights on a small amount of data (App. F.8) and that our attack shows remarkable transferability across datasets and data distributions (Sec. 5 and App. F.4 and App. F.11).

## C   RECONSTRUCTING FROM SECURELY-AGGREGATED GRADIENT UPDATES

As described in Sec. 4.1, we have designed a more elaborate property $\mathcal{P}$ for the case of attacking securely aggregated gradient updates. Our property is based on a combination of local (in-batch) and global distribution information about the client data allowing us to handle this more complex case. In this section, we describe in detail how this is done.

As described in Sec. 4.1, we need to define the property $\mathcal{P}$ with respect to a range of brightnesses, such that $P(|I_{\text{rec}}| = 1)$ is maximized. Thus, generating $\mathcal{P}$ is reduced to finding the correct threshold $\tau$ on the client image brightnesses, with which we later train SEER.

To calculate the threshold $\tau$ in the multi-client setting, we use the insight that individual client batches are still generated in the presence of BN before their aggregation. To this end, we normalize the brightnesses within individual client batches of size $B$ for 20000 sampled client batches and use the sampled normalized brightness to generate the cumulative density function(CDF) of their empirical distribution. We then choose the threshold $\tau$ on this distribution to maximize the probability that exactly one out of the $C$ aggregated clients has exactly one image with normalized brightness above the threshold. For simplicity, we demonstrate this in the case of maximal brightness, as the minimal-brightness case is equivalent. We estimate the probability of having exactly one image with normalized brightness above the threshold as:

$$(1 - \Phi_1(\tau)) * \Phi_2(\tau) * \Phi_1(\tau)^{C-1} \tag{6}$$

where $\Phi_1$ is the CDF of the top brightness in a sampled batch, and $\Phi_2$ is the CDF of the second-highest brightness in a sampled batch. The equation can be intuitively rephrased as follows—for exactly one client, the highest normalized brightness within its batch is above $\tau$, *and* the second-highest brightness is below $\tau$, while for the rest, *all* brightnesses are below $\tau$. To optimize Eq. 6 for the threshold $\tau$, we use the golden section search method - a numerical optimization technique that repeatedly divides a search interval by the golden ratio to efficiently locate the (possibly-local) extremum of a function of a single variable.

## D   EXISTENCE OF THE PROPERTY $\mathcal{P}$

A key assumption of our attack is the existence of an image property $\mathcal{P}$ satisfied by exactly one image $\boldsymbol{x} \in \mathbb{R}^d$ in the client batch. We note that if we do not impose any restrictions on $\mathcal{P}$, properties of the type "equal to $\boldsymbol{x}$" satisfy the requirement as they can single out any $\boldsymbol{x}$ from any batch. However, such properties do not generalize well across batches, limiting SEER's practical applicability. A more interesting question then becomes, is there always a property $\mathcal{P}$ that can separate exactly one image from a client batch that additionally is also (i) simple enough to train SEER's malicious weights on well, and (ii) transferable across batches without retraining. To this end, in this section we theoretically investigate the existence of properties $\mathcal{P}$ of the type $m(\boldsymbol{x}) < \tau$, where $m$ is a linear function and $\tau \in \mathbb{R}$ is a threshold, as in our experiments (see App. F.6) these types of properties show good learnability and generalizability.

Under these assumptions, we can reformulate the question if $\mathcal{P}$ exists for a given batch and chosen image $\boldsymbol{x}$ in it, as the question if a hyperplane $m(\boldsymbol{x}) - \tau = 0$ exists that separates the chosen image, represented as a point in $d$-dimensional space, from the rest of the images in the batch. This question is answered positively by a well-known result from theoretical ML (see e.g., Lauer [2017]) which states that the VC dimension of affine classifiers (in this case $sgn(m(\boldsymbol{x}) - \tau)$) in $\mathbb{R}^d$ is $d + 1$. Thus, a linear property $\mathcal{P}$ that is true only for the chosen images $\boldsymbol{x}$ in the batch always exists when the batch

Table 5: Effect of gradient clipping when different maximum gradient norms $\mathcal{C}$ are applied on CIFAR10 model trained with the Red property on $B = 128$. We report the percentage of well-reconstructed images (*Rec*), the average PSNR and its standard deviation across all reconstructions (*PSNR-All*), and on the top $37\%$ images (*PSNR-Top*).

| $\mathcal{C}$ | Rec (%) | PSNR-Top ↑ | PSNR-All ↑ |
|---|---|---|---|
| 1.0 | 36.2 | $21.5 \pm 1.7$ | $17.8 \pm 3.4$ |
| 2.0 | 87.3 | $25.5 \pm 1.4$ | $22.3 \pm 3.2$ |
| 3.0 | 92.6 | $27.7 \pm 1.1$ | $24.7 \pm 3.3$ |
| 4.0 | **95.2** | $28.7 \pm 1.0$ | $25.9 \pm 3.2$ |
| $\infty$ | 93.5 | $\mathbf{31.1 \pm 1.2}$ | $\mathbf{27.8 \pm 4.1}$ |

Table 6: Effect of applying DP-SGD with maximum gradient norm of $\mathcal{C} = 3$ and different noise levels $\sigma$ on CIFAR10 model trained with the Red property on $B = 128$. We report the percentage of well-reconstructed images (*Rec*), the average PSNR and its standard deviation across all reconstructions (*PSNR-All*), and on the top $37\%$ images (*PSNR-Top*).

| $\sigma$ | Rec (%) | PSNR-Top ↑ | PSNR-All ↑ |
|---|---|---|---|
| 0.0 | **92.6** | $\mathbf{27.7 \pm 1.1}$ | $\mathbf{24.7 \pm 3.3}$ |
| $1 \times 10^{-4}$ | **92.6** | $\mathbf{27.7 \pm 1.1}$ | $\mathbf{24.7 \pm 3.3}$ |
| $1 \times 10^{-3}$ | 92.4 | $27.2 \pm 1.0$ | $24.4 \pm 3.1$ |
| $3 \times 10^{-3}$ | 90.3 | $24.6 \pm 0.6$ | $22.6 \pm 2.4$ |
| $5 \times 10^{-3}$ | 84.0 | $21.9 \pm 0.4$ | $20.4 \pm 1.9$ |
| $7 \times 10^{-3}$ | 43.5 | $19.7 \pm 0.4$ | $18.4 \pm 1.8$ |
| $1 \times 10^{-2}$ | 0.0 | $17.2 \pm 0.4$ | $15.3 \pm 2.4$ |

images are in a general position (which usually holds) and $B \leq d + 1$ (which holds even for low-dim images like CIFAR10, as $d + 1 = 3073$ which is far above practical batch sizes).

# E  POSSIBLE DEFENSES AGAINST SEER

Here we discuss other possible defenses against SEER, and why we believe they are not currently effective at preventing data leakage. Detection based on tracking the value of the loss during training is, as noted in Boenisch et al. [2021], unlikely to work. This is due to the fact that in FL, per-client values can generally be noisy, even if the global loss consistently falls. Such detection is also made harder by the fact that SEER does not require application in more than one round to pose a serious threat to client privacy, and that such attacks are often applied in the first round [Balunovic et al., 2022b]. Next, SEER can't be easily flagged via other kinds of client-side detection, as it only modifies the weights through continuous optimization, using no obvious handcrafted patterns (as opposed to prior work analyzed in the paper). Finally, defenses based on differential privacy such as DP-SGD [Abadi et al., 2016] require too much noise to be practical, as we also demonstrate in our experiments in App. E.1. As we note in Sec. 6, we think that future principled client-side defenses can be a promising future direction.

## E.1  RESULTS UNDER DIFFERENTIAL PRIVACY

We demonstrate SEER's performance under defenses based on differential privacy. In particular, we apply our attack on gradients obtained from the DP-SGD [Abadi et al., 2016] algorithm. In DP-SGD, the clients defend their data by first clipping the norms of the per-layer gradients of each of their data points to at most $\mathcal{C}$, and then adding Gaussian noise with standard deviation of $\mathcal{C} \cdot \sigma$ to them. In order to better understand what effect those two components of the defense have on our method, in Table 5 we experiment with different clipping norms $\mathcal{C}$ for $\sigma = 0$ (i.e., no noise added), while in Table 6 we experiment with the noise strength $\sigma$ for clipping norm of 3. We use $\mathcal{C} = 3$ as Abadi et al. [2016] recommends that value for CIFAR10, the dataset we experiment with. Both experiments were conducted on a SEER model trained with $B = 128$ and the Red property.

Table 7: Large batch reconstruction on bright and red properties from batches of different sizes $B$ on CIFAR10. We report the percentage of well-reconstructed images (*Rec*), the average PSNR and its standard deviation on all reconstructions (*PSNR-All*), and across the top $37\%$ images (*PSNR-Top*).

| | CIFAR10, Red | | | CIFAR10, Bright | | |
|---|---|---|---|---|---|---|
| $B$ | Rec (%) | PSNR-Top $\uparrow$ | PSNR-All $\uparrow$ | Rec (%) | PSNR-Top $\uparrow$ | PSNR-All $\uparrow$ |
| 64 | 87.3 | $30.4 \pm 1.1$ | $26.5 \pm 5.2$ | 89.4 | $32.1 \pm 2.0$ | $27.2 \pm 5.3$ |
| 128 | 93.5 | $31.1 \pm 1.2$ | $27.8 \pm 4.1$ | **94.2** | $31.9 \pm 1.7$ | $28.2 \pm 4.3$ |
| 256 | **94.7** | $\mathbf{31.3 \pm 1.0}$ | $\mathbf{28.0 \pm 4.0}$ | 93.5 | $\mathbf{32.8 \pm 2.0}$ | $\mathbf{28.5 \pm 5.0}$ |
| 512 | 94.4 | $30.0 \pm 1.2$ | $26.6 \pm 3.8$ | 87.8 | $26.6 \pm 1.8$ | $23.2 \pm 3.5$ |

Table 8: Large batch reconstruction on bright and red properties from batches of different sizes $B$ on CIFAR100. We report the percentage of well-reconstructed images (*Rec*), the average PSNR and its standard deviation across all reconstructions (*PSNR-All*), and on the top $37\%$ images (*PSNR-Top*).

| | CIFAR100, Red | | | CIFAR100, Bright | | |
|---|---|---|---|---|---|---|
| $B$ | Rec (%) | PSNR-Top $\uparrow$ | PSNR-All $\uparrow$ | Rec (%) | PSNR-Top $\uparrow$ | PSNR-All $\uparrow$ |
| 64 | 97.1 | $31.7 \pm 1.1$ | $29.0 \pm 3.4$ | 95.6 | $32.2 \pm 1.5$ | $28.2 \pm 4.3$ |
| 128 | 97.4 | $31.8 \pm 1.1$ | $29.3 \pm 3.2$ | 94.7 | $30.0 \pm 1.3$ | $26.5 \pm 3.7$ |
| 256 | 97.7 | $31.3 \pm 1.0$ | $28.6 \pm 3.2$ | **98.1** | $\mathbf{35.2 \pm 1.4}$ | $\mathbf{30.8 \pm 4.8}$ |
| 512 | **98.6** | $\mathbf{33.1 \pm 1.1}$ | $\mathbf{30.5 \pm 3.1}$ | 95.0 | $32.2 \pm 1.6$ | $27.6 \pm 4.6$ |

In our experiments, the performance of SEER increased when we explicitly took into account the use of DP-SGD during our attack procedure. In particular, before applying the attack in all experiments we first estimate the median clipping factor for each layer individually on 1000 batches of size 128 taken from our auxiliary data. These approximate factors are then reapplied to the total gradients sent from the clients to the malicious server before applying Algorithm 2.

The results in Table 5 and Table 6 confirm the trends observed in prior work [Zhu et al., 2019; Geiping et al., 2020; Balunovic et al., 2022b] that low clipping norms $\mathcal{C}$ and high noise levels $\sigma$ result in more effective defense mechanisms. Still, we observe that SEER is fairly robust to DP-SGD, as even for clipping norm of 2, and high noise levels of $5 \times 10^{-3}$ our method is able to recover private date from more than $85\%$ of client batches.

## F   ADDITIONAL EXPERIMENTS

In this section, we provide additional experimental results, which we did not include in the main body due to space constraints.

### F.1   EXTENDED LARGE BATCH EXPERIMENTS

In Table 7 and Table 8, we present the extended version of our CIFAR10 and CIFAR100 single-client experiments, first presented in Sec. 5. We observe similar trends as in the original experiments. For example, we observe that CIFAR10 and CIFAR100 performances are similar and that there is no major difference in performance between the most bright and most red image properties.

### F.2   EXTENDED SECURE AGGREGATION EXPERIMENTS

In Table 9, we present an extended version of our CIFAR10 multi-client experiments first presented in Sec. 5 containing results for both the bright and dark properties. While we see the dark property reconstructions are generally slightly better than the bright ones, we observe similar trends between the two sets of the experiments. This suggests our method for attacking secure aggregation federated updates is effective regardless the property used.

Table 9: Reconstructions on securely aggregated batches on the bright and dark properties with different numbers of clients $C$ on CIFAR10, for different total numbers of images. We report the percentage of correctly reconstructed images (*Rec*) and the average PSNR across the top 37% images (*PSNR-Top*).

| #Imgs | $C = 4$, Dark | | $C = 4$, Bright | | $C = 8$, Dark | | $C = 8$, Bright | |
|---|---|---|---|---|---|---|---|---|
| | Rec (%) | PSNR-Top | Rec (%) | PSNR-Top | Rec (%) | PSNR-Top | Rec (%) | PSNR-Top |
| 64 | 50.2 | $27.5 \pm 3.0$ | 41.4 | $\mathbf{27.3 \pm 3.1}$ | 43.0 | $27.4 \pm 3.3$ | 41.3 | $26.6 \pm 3.7$ |
| 128 | 51.3 | $28.8 \pm 2.6$ | 44.2 | $26.8 \pm 3.0$ | 43.4 | $27.6 \pm 3.5$ | 40.6 | $\mathbf{27.3 \pm 3.3}$ |
| 256 | 50.9 | $29.8 \pm 2.3$ | 51.9 | $\mathbf{27.3 \pm 2.5}$ | 51.7 | $27.0 \pm 2.9$ | 41.9 | $25.4 \pm 3.1$ |
| 512 | $\mathbf{61.3}$ | $\mathbf{30.2 \pm 2.4}$ | $\mathbf{52.9}$ | $25.7 \pm 2.4$ | $\mathbf{56.3}$ | $\mathbf{28.7 \pm 2.9}$ | $\mathbf{51.7}$ | $25.9 \pm 2.8$ |

Table 10: Large batch reconstruction on the bright property on CIFAR10 on a network trained with batch size $B = 128$ and tested for various client batch sizes $B_{\text{test}}$. We report the percentage of well-reconstructed images (*Rec*), the average PSNR and its standard deviation on all reconstructions (*PSNR-All*), and across the top 37% images (*PSNR-Top*).

| $B_{\text{test}}$ | Rec (%) | PSNR-Top $\uparrow$ | PSNR-All $\uparrow$ |
|---|---|---|---|
| 64 | 42.0% | $21.51 \pm 1.83$ | $18.51 \pm 2.93$ |
| 96 | 87.4% | $30.56 \pm 1.31$ | $26.28 \pm 4.92$ |
| 128 | $\mathbf{94.2\%}$ | $\mathbf{31.91 \pm 1.73}$ | $\mathbf{28.15 \pm 4.34}$ |
| 192 | 86.3% | $30.78 \pm 2.37$ | $25.77 \pm 5.14$ |
| 256 | 67.5% | $28.02 \pm 2.79$ | $22.42 \pm 5.14$ |

## F.3 ROBUSTNESS TO $B$

In this section, we demonstrate that attack parameters $\theta_f$ generated by SEER for a particular client batch size $B$ can work to a large extent for batch sizes close to the original one, thus relaxing the requirement that the exact client batch size $B$ is known during the crafting of the malicious model $f$. In particular, in Table 10, we show the effect of applying our single-client attack trained on $B = 128$ on CIFAR10 using the *Bright* image property for clients with varying batch sizes $B_{\text{test}}$. We observe that while, as expected, SEER performs best when $B_{\text{test}} = B$, both the success rate and the quality of reconstruction on clients with batch sizes even $2\times$ larger than the trained one remain very good. We note that Table 10 suggests that underestimating the client batch size $B_{\text{test}}$ during the training of $f$ is better than overestimating it, as the reconstruction performance when $B_{\text{test}} = 64$ is significantly worse than when $B_{\text{test}} = 256$. This is mostly caused by $d$ filtering out all images in the client batches resulting in the removal of all the client data.

## F.4 ROBUSTNESS TO IMAGE CORRUPTIONS

In this section, we show that SEER is robust to distributional shifts caused by common image corruptions. To this end, we use a SEER model trained on the CIFAR10 trainset with the red property, batch size $B = 128$, and without secure aggregation to attack batches sampled from the CIFAR10-C dataset [Hendrycks & Dietterich, 2018], where 19 different image corruptions are applied at different levels of severity (1-5) to the original testset of CIFAR10. We show the results in Table 11.

We see that for all image corruptions but *Fog* and *Contrast*, even at severity 5, we recover images from more than 85% of client batches with good quality (average PSNR > 23 in most cases). For *Fog* and, to even greater extend, *Contrast*, however, we observed that reconstructions, while preserving the image semantic, became too bright resulting in a very low PSNR numbers. To this end, we created a modified version of our attack that approximates and applies a multiplicative factor $\beta$ by which one needs to multiply the recovered normalized images such that 90% of the recovered images after denormalization are inside the range $[0, 1]$. We use 90% to account for the fact that not all images will be correctly recovered, and we don't want these to affect our $\beta$ estimations. The results are shown under *Fog Fixed* and *Contrast Fixed*, where we get even better results than on the original data.

Table 11: Large reconstruction in the presence of different common corruptions applied on CIFAR10 network trained on $B = 128$ using the red property. As a baseline, our attack achieves Rec: $93.5\%$, PSNR-Top: $31.1 \pm 1.2$, and PSNR-All: $27.8 \pm 4.1$ . We report the percentage of well-reconstructed images (*Rec*), the average PSNR and its standard deviation across all reconstructions (*PSNR-All*), and on the top $37\%$ images (*PSNR-Top*) for three different degrees of severity of the corruption (*Severity*).

| | Severity = 1 | | | Severity = 3 | | | Severity = 5 | | |
|---|---|---|---|---|---|---|---|---|---|
| Corruption | Rec (%) | PSNR-Top ↑ | PSNR-All ↑ | Rec (%) | PSNR-Top ↑ | PSNR-All ↑ | Rec (%) | PSNR-Top ↑ | PSNR-All ↑ |
| Brightness | 94.7 | **31.4 ± 1.1** | **28.0 ± 4.1** | 92.3 | 31.6 ± 1.1 | 27.7 ± 4.5 | 94.0 | **31.0 ± 1.4** | 27.3 ± 4.3 |
| Contrast | **96.2** | 24.6 ± 1.0 | 22.9 ± 2.0 | 2.8 | 17.8 ± 0.9 | 16.6 ± 1.2 | 0.6 | 15.1 ± 1.3 | 13.6 ± 1.6 |
| Contrast Fixed | **97.5** | 34.4 ± 1.3 | 30.6 ± 4.5 | 99.0 | 32.2 ± 1.5 | 29.2 ± 3.29 | 98.0 | 28.2 ± 1.6 | 25.4 ± 2.9 |
| Defocus Blur | 94.2 | 30.2 ± 1.1 | 27.3 ± 3.7 | 94.9 | 27.5 ± 1.1 | 25.1 ± 2.9 | 94.8 | 25.2 ± 0.9 | 23.2 ± 2.3 |
| Elastic Transform | 95.0 | 29.0 ± 1.0 | 26.3 ± 3.3 | 94.7 | 27.7 ± 1.1 | 25.2 ± 2.9 | 95.3 | 28.1 ± 1.1 | 25.5 ± 3.0 |
| Fog | 95.3 | 26.1 ± 1.1 | 24.0 ± 2.3 | 68.6 | 21.4 ± 1.1 | 19.8 ± 1.6 | 23.0 | 19.5 ± 0.9 | 18.1 ± 1.3 |
| Fog Fixed | 95.4 | 33.6 ± 1.4 | 29.4 ± 4.7 | 98.0 | 34.6 ± 1.2 | 31.0 ± 4.4 | 97.6 | 35.0 ± 1.0 | 31.1 ± 4.7 |
| Frost | 94.4 | 30.2 ± 0.9 | 27.0 ± 3.8 | 94.6 | 28.1 ± 1.0 | 25.4 ± 3.3 | 94.2 | 26.5 ± 0.7 | 24.3 ± 2.6 |
| Gaussian Blur | 94.2 | 30.3 ± 1.1 | 27.3 ± 3.7 | 95.0 | 26.5 ± 1.0 | 24.3 ± 2.6 | 95.0 | 24.3 ± 1.0 | 22.5 ± 2.1 |
| Gaussian Noise | 94.2 | 30.3 ± 0.8 | 27.2 ± 3.9 | 92.8 | 27.8 ± 0.5 | 25.2 ± 3.4 | 90.9 | 26.3 ± 0.4 | 24.0 ± 3.1 |
| Glass Blur | 93.3 | 31.2 ± 1.3 | 27.7 ± 4.2 | 95.0 | 29.6 ± 1.1 | 26.8 ± 3.5 | 94.9 | 29.8 ± 1.1 | 27.0 ± 3.4 |
| Impulse Noise | 93.0 | 29.5 ± 0.8 | 26.6 ± 3.7 | 90.9 | 26.4 ± 0.7 | 24.0 ± 3.0 | 85.8 | 22.5 ± 0.5 | 20.9 ± 2.1 |
| Jpeg Compression | 94.6 | 30.9 ± 1.1 | 27.7 ± 3.9 | **95.3** | 30.8 ± 1.1 | 27.7 ± 3.8 | 94.5 | 30.6 ± 1.1 | **27.6 ± 3.8** |
| Motion Blur | 95.3 | 28.6 ± 1.0 | 26.1 ± 3.1 | 95.0 | 26.2 ± 1.1 | 24.1 ± 2.5 | 94.6 | 25.3 ± 1.2 | 23.3 ± 2.3 |
| Pixelate | 93.3 | 30.8 ± 1.1 | 27.6 ± 3.9 | 94.2 | 30.4 ± 1.1 | 27.3 ± 3.8 | 94.4 | 29.1 ± 1.0 | 26.5 ± 3.4 |
| Saturate | 93.5 | 29.1 ± 1.2 | 26.1 ± 3.6 | 95.0 | **31.7 ± 1.5** | **28.0 ± 4.2** | 91.4 | 24.7 ± 0.8 | 22.7 ± 2.3 |
| Shot Noise | 94.2 | 30.7 ± 0.9 | 27.6 ± 3.9 | 92.9 | 29.0 ± 0.7 | 26.2 ± 3.6 | 91.5 | 27.0 ± 0.6 | 24.6 ± 3.3 |
| Snow | 93.3 | 31.3 ± 1.1 | 27.9 ± 4.2 | 92.6 | 30.8 ± 0.9 | 27.3 ± 4.2 | 91.8 | 29.3 ± 1.1 | 25.9 ± 4.0 |
| Spatter | 93.3 | 31.3 ± 1.1 | 27.9 ± 4.2 | 92.5 | 31.1 ± 1.0 | 27.5 ± 4.4 | 93.0 | 30.6 ± 1.1 | 27.1 ± 4.2 |
| Speckle Noise | 93.8 | 30.8 ± 1.0 | 27.6 ± 3.9 | 92.5 | 29.5 ± 0.8 | 26.5 ± 3.9 | 91.2 | 27.0 ± 0.8 | 24.5 ± 3.3 |
| Zoom Blur | 93.6 | 27.9 ± 1.1 | 25.3 ± 3.2 | 93.6 | 26.6 ± 1.0 | 24.3 ± 2.8 | **95.4** | 25.5 ± 1.0 | 23.5 ± 2.3 |

Table 12: Large batch reconstruction on the bright and dark properties from batches of size $B = 128$ on CIFAR10 using local (*Local*) and global properties $\mathcal{P}$ (*Global*). We report the percentage of well-reconstructed images (*Rec*), the average PSNR and its standard deviation across all reconstructions (*PSNR-All*), and across the top $37\%$ images (*PSNR-Top*).

| | CIFAR10, Bright | | | CIFAR10, Dark | | |
|---|---|---|---|---|---|---|
| $\mathcal{P}$ | Rec (%) | PSNR-Top ↑ | PSNR-All ↑ | Rec (%) | PSNR-Top ↑ | PSNR-All ↑ |
| Global | 54.4 | 27.0 ± 1.8 | 20.6 ± 5.8 | 61.9 | 27.7 ± 2.2 | 21.1 ± 6.1 |
| Local | **94.2** | **31.9 ± 1.7** | **28.2 ± 4.3** | **81.3** | **33.6 ± 1.4** | **27.4 ± 7.3** |

We conjecture this is due to the fact that most of the corrupted images have narrower range of pixel values making obtaining high PSNR numbers easier. All in all, our experiments show that SEER is very robust to image corruptions, and that even the most severe changes like the ones in CIFAR10-C, sometimes resulting in hard to recognize images, can be handled well by our algorithm.

## F.5 COMPARISON BETWEEN OUR LARGE BATCH AND SECURE AGGREGATION PROPERTIES

In this section, we compare our two SEER variants—one based purely on the local distribution of $m$ for the large batch setting, referred to as local $\mathcal{P}$, and the other for secure aggregation setting, described in App. C, which is based on a mix of the global and local distributions of $m$ and is referred to as global $\mathcal{P}$. We mount both variants of SEER on gradients coming from a single client with batch size $B = 128$ on CIFAR10. We note that both methods are well-defined in this setting, and either one can successfully reconstruct data from the client batches.

The results are depicted in Table 12. While both methods successfully reconstruct the majority of client batches, we clearly see the benefits of using the local property $\mathcal{P}$. In particular, the results in Table 12 suggest that the local distribution approach reconstructs up to $1.75$ times more images, while also producing higher PSNR values not only on the full set of reconstructed batches but also on the top $37\%$ of them. This motivates the need for our single-client attack variant, and demonstrates that secure aggregation provides additional protection to individual clients.

Table 13: Large batch reconstruction on CIFAR10 with $B = 128$ with different properties $\mathcal{P}$. We report the percentage of well-reconstructed images (*Rec*), the average PSNR and its standard deviation on all reconstructions (*PSNR-All*), and across the top $37\%$ images (*PSNR-Top*).

| Property | Rec (%) | PSNR-Top ↑ | PSNR-All ↑ |
|---|---|---|---|
| Bright | 94.2 | $31.9 \pm 1.7$ | $28.2 \pm 4.3$ |
| Dark | 81.3 | $\mathbf{33.6 \pm 1.4}$ | $27.4 \pm 7.3$ |
| Red | 93.5 | $31.1 \pm 1.2$ | $27.8 \pm 4.1$ |
| Blue | 97.2 | $31.5 \pm 0.9$ | $28.6 \pm 3.5$ |
| Green | 96.7 | $32.8 \pm 1.1$ | $\mathbf{29.6 \pm 4.0}$ |
| H Edge | 80.1 | $29.0 \pm 1.1$ | $24.4 \pm 5.5$ |
| V Edge | 85.8 | $29.6 \pm 1.0$ | $25.5 \pm 5.0$ |
| Green V Edge | 95.1 | $32.5 \pm 1.1$ | $28.6 \pm 4.5$ |
| Rand | $\mathbf{97.5}$ | $32.8 \pm 1.1$ | $29.4 \pm 3.8$ |

## F.6 ADDITIONAL TYPES OF PROPERTY METRICS $m$

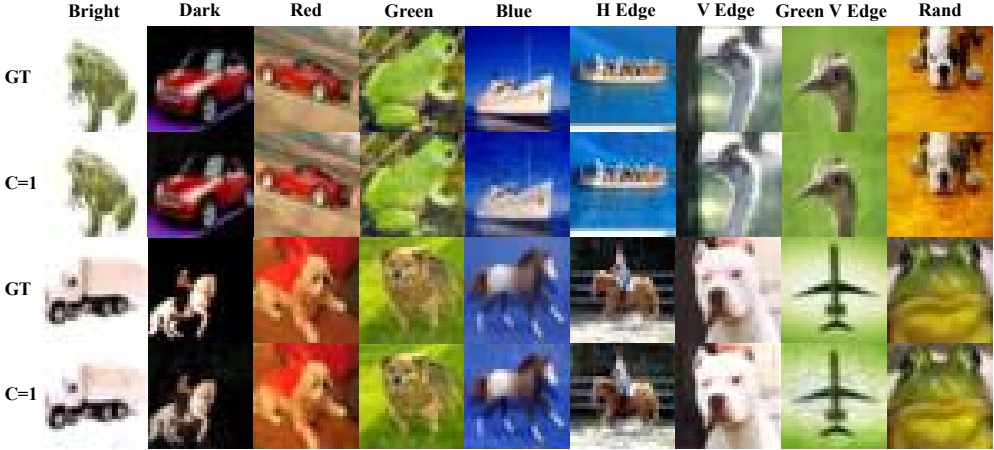

Figure 5: Example reconstructions of SEER trained on CIFAR10 with $C = 1, B = 128$ and different properties.

In this section, we demonstrate that our method works well for variety of properties $\mathcal{P}$ based on different metrics $m$. In particular, we look at local properties $\mathcal{P}$ based on: (i) the image brightness—the most bright (*Bright*) and the most dark (*Dark*) image in a batch; (ii) the image color—the most red (*Red*), blue (*Blue*), or green (*Green*) image in a batch; (iii) edges in the image—the image with the strongest horizontal (*H Edge*), or vertical (*V Edge*) edges in a batch; (iv) combination of image color and edges—the most green image with vertical edges (*Green V Edge*); and, finally, (iv) based on random property (*Rand*). For the color properties we rank the batch images based on the difference between two times the average color channel response for the chosen color and the sum of the other two average color channel responses. For the edge properties, we ranked the batch images based on the average response with the $[1, -1]$ edge filter (either in horizontal or vertical direction) on a grayscale version of the image. Further, for the combination filter we ranked images based on sum of the color and edge property scores. Finally, for the random property we ranked the batch images based on the average response to a random $3 \times 3$ convolution filter that was normalized. The results for CIFAR10 for the large batch size setting for $B = 128$ is shown in Table 13. Further, example reconstructions are given in Fig. 5.

We observe that for all properties SEER successfully recovers data from a large portion of the client batches (>80%), with good quality (PSNR>24). Yet, we still observe some variability across the properties with the Random property being the easiest to attack, as demonstrated by the percentage of recovered images and the very good PSNR metrics. This is in line with the observations in Fowl et al. [2022b]. We also observed that the Dark property produces the best image reconstructions, as shown

Table 14: Large batch reconstruction on CIFAR10 model trained with the Bright property on $B = 128$ after several rounds of federated training (*#Rounds*) with different learning rates (*Learning Rate*) using 8 clients. We report the percentage of well-reconstructed images (*Rec*), the average PSNR and its standard deviation across all reconstructions (*PSNR-All*), and on the top $37\%$ images (*PSNR-Top*).

| #Rounds | Learning Rate | Rec (%) | PSNR-Top ↑ | PSNR-All ↑ |
|---|---|---|---|---|
| 0 | $1 \times 10^{-4}$ | 94.2 | $\mathbf{31.9 \pm 1.7}$ | $\mathbf{28.2 \pm 4.3}$ |
| 1 | $1 \times 10^{-4}$ | **94.6** | $31.8 \pm 1.7$ | $\mathbf{28.2 \pm 4.3}$ |
| 2 | $1 \times 10^{-4}$ | 94.4 | $31.1 \pm 1.6$ | $27.3 \pm 4.2$ |
| 3 | $1 \times 10^{-4}$ | 89.2 | $27.2 \pm 1.7$ | $23.6 \pm 3.6$ |
| 4 | $1 \times 10^{-4}$ | 25.8 | $20.1 \pm 1.5$ | $17.7 \pm 2.3$ |
| 5 | $1 \times 10^{-4}$ | 6.1 | $17.5 \pm 1.5$ | $14.9 \pm 2.3$ |
| 1 | $5 \times 10^{-4}$ | 91.4 | $29.0 \pm 1.7$ | $25.1 \pm 4.0$ |

by PSNR-Top, but fails to reconstruct as often. In practice, we observed this happens due to SEER recovering completely black images. We conjecture this is due to lack of diversity in the images SEER sees during training as most images in CIFAR10 with the Dark property have completely black background. Finally, we observe that color properties are easier to attack compared to edge ones but, interestingly, when combined, like in *Green V Edge*, the results become much closer to the color version. We conjecture this is due images with very pronounced colors being more similar to each other and, thus, easier to distinguish from the rest of the images in the batch.

## F.7 RESULTS UNDER TRAINED ENCODER

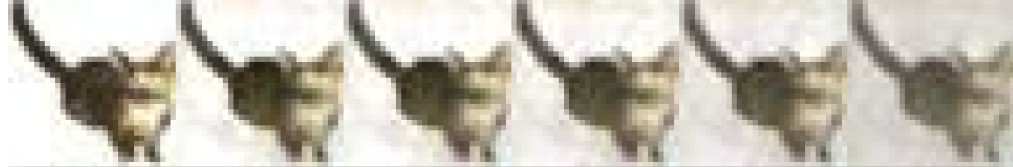

Figure 6: Example reconstructions of SEER after training the client model for different number of FL rounds using 8 clients without secure aggregation with learning rate $1 \times 10^{-4}$. Left to right: Ground truth, 1 round, 2 rounds, 3 rounds, 4 rounds, 5 rounds.

Throughout this paper, we assumed that our attack is mounted at the beginning of the training procedure. In this section instead, we show the results of using SEER's decoder to reconstruct images from gradients computed on models that were trained for several federated learning rounds from the malicious state chosen by SEER. We show the results quantitatively in Table 14 and qualitatively in Fig. 6 for different number of rounds (*#Rounds*) and learning rates *Learning Rate* when training with 8 clients without using secure aggregation.

We observe that our decoder works even after several rounds of training of the client model. Further, we observe that our decoder is more robust to a single large step update (One round with learning rate $5 \times 10^{-4}$) than many smaller updates applied sequentially (Five rounds with learning rate $1 \times 10^{-4}$) and that each additional communication round results in a small additional drop in quality of the reconstruction. We believe further improvements over these results are achievable if one finetunes our decoder model at each FL round to match the changes of the client encoder, but we leave this as future work.

## F.8 EXPERIMENTS WITH AUXILIARY DATASETS OF DIFFERENT SIZES

In this section we investigate how the size of the auxiliary dataset used for training SEER affects its results. In particular, in Table 15, we show the results when we used only $p$ percent of the data points in CIFAR10's train set as auxiliary data. As expected of any algorithm based on training, our results generally improve with the number of datapoints available to the attacker. Despite this, we see that our method is very sample efficient, as we successfully reconstruct data from $> 80\%$ of

Table 15: Reconstructions on models trained with the Red property and $B = 128$ on different percentage $p$ of data in the CIFAR10 trainset. We report the percentage of well-reconstructed images (*Rec*), the average PSNR and its standard deviation across all reconstructions (*PSNR-All*), and on the top $37\%$ images (*PSNR-Top*).

| $p$ | Rec (%) | PSNR-Top ↑ | PSNR-All ↑ |
|---|---|---|---|
| 5 | 80.6 | $24.5 \pm 1.6$ | $21.5 \pm 3.1$ |
| 10 | 92.2 | $27.9 \pm 1.1$ | $24.7 \pm 3.4$ |
| 20 | 86.9 | $30.2 \pm 0.9$ | $26.4 \pm 4.9$ |
| 33 | 87.5 | $\mathbf{31.7 \pm 1.1}$ | $27.2 \pm 5.5$ |
| 50 | $\mathbf{95.8}$ | $31.5 \pm 1.1$ | $\mathbf{28.3 \pm 3.8}$ |
| 100 | 93.5 | $31.1 \pm 1.2$ | $27.8 \pm 4.1$ |

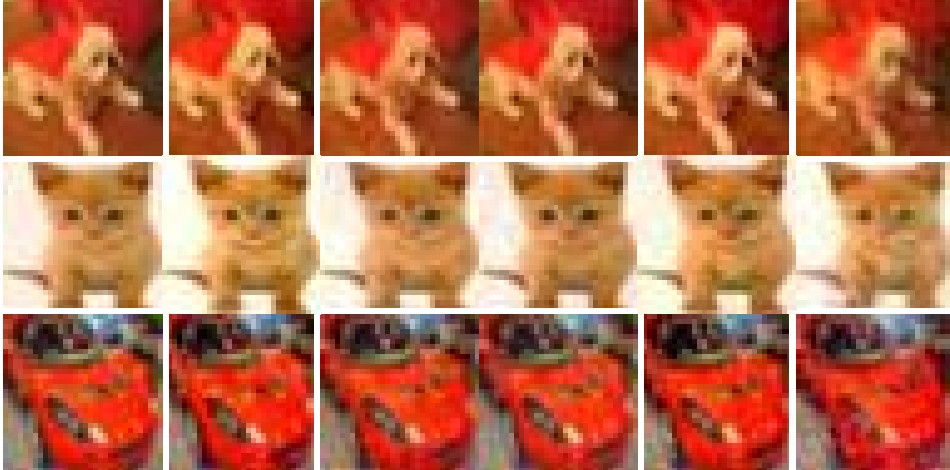

Figure 7: Example reconstructions of SEER trained on different percentage $p$ of CIFAR10 train set. Left to right: Ground truth, 100%, 50%, 33%, 20%, 10%, and 5%.

client batches even when training on mere 2500 training data samples ($p = 5\%$). We further show qualitative comparison between the models in Fig. 7, where we confirm the quality of reconstructions degrades when less data is available, especially for very small $p$, yet for all $p$ the images remain recognizable regardless.

### F.9    RESULTS UNDER CLIENT HETEROGENEITY

Next, we look at how our method is affected by the level of client data heterogeneity. In particular, we take our CIFAR10 model trained with $B = 128$ and the Red property, and we evaluate its performance on clients with different level of non-IID data. To simulate non-IID data, we sample from a Dirichlet distribution with parameter $\alpha$, to determine each client's label distribution and randomly sample the client data according to this. In this setting, $\alpha$ close to $0$ means that each client only holds data from few classes. We show the results in Table 16, where we show that while heterogeneity slightly degrades our performance, SEER is very robust to heterogeneity, as even severe heterogeneity levels like $\alpha = 0.1$ produce above $90\%$ attack success rate with average PSNR $> 25$.

### F.10    RESULTS ON THE RESIMAGENET DATASET

In this section, we show quantitative and qualitative results from applying SEER on the ResImageNet dataset (Restricted ImageNet [Tsipras et al., 2018], a subset of ImageNet with 9 superclasses). Our setup is similar to the experiments presented for ImageNet in the main paper, i.e., we use batch size $B = 64$, the Bright property, and our U-Net decoder architecture (App. G). Further, we also use the same hyperparameters (App. H), except for two small changes: (i) we execute the pretraining stage on the images in CIFAR10, instead of the downsized version of the images in ImageNet, avoiding

Table 16: Reconstructions for clients with different data heterogeneity levels $\alpha$ on a CIFAR10 model trained with the Red property on $B = 128$. We report the percentage of well-reconstructed images (*Rec*), the average PSNR and its standard deviation across all reconstructions (*PSNR-All*), and on the top $37\%$ images (*PSNR-Top*).

| $\alpha$ | Rec (%) | PSNR-Top $\uparrow$ | PSNR-All $\uparrow$ |
|---|---|---|---|
| 0.1 | 93.9 | $28.1 \pm 1.0$ | $25.3 \pm 3.3$ |
| 0.2 | 94.4 | $28.7 \pm 1.1$ | $26.0 \pm 3.3$ |
| 0.3 | 94.3 | $29.1 \pm 1.0$ | $26.2 \pm 3.4$ |
| 0.4 | 95.1 | $29.2 \pm 1.1$ | $26.4 \pm 3.4$ |
| 0.5 | 95.0 | $29.4 \pm 1.0$ | $26.6 \pm 3.4$ |
| 0.6 | **96.2** | $29.5 \pm 1.0$ | $26.7 \pm 3.2$ |
| 0.7 | 95.8 | $29.4 \pm 1.1$ | $26.6 \pm 3.3$ |
| 0.8 | 95.8 | $29.7 \pm 1.1$ | $\mathbf{27.0 \pm 3.3}$ |
| 0.9 | 95.8 | $29.6 \pm 1.0$ | $26.9 \pm 3.3$ |
| 1.0 | 96.1 | $\mathbf{29.8 \pm 1.1}$ | $\mathbf{27.0 \pm 3.3}$ |

**Ground Truth**

**C=1**

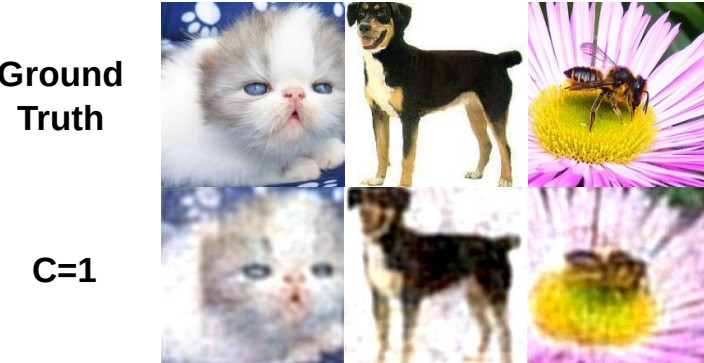

Figure 8: Example reconstructions of SEER on ResImageNet for $B = 64$ and the Bright property.

pretraining on the full 1000 classes, (ii) we train for 370 epochs instead, with 400 gradient descent steps per epoch.

Under these settings, SEER is able to recover $77\%$ of images with average PSNR of $20.6 \pm 3.7$ and PSNR Top of $23.8 \pm 1.4$. Examples are shown in Fig. 8, where we see that images are clearly recognizable and accurate in terms of object positions.

### F.11    RESULTS UNDER DATA DOMAIN SHIFTS

**Ground Truth**

**C=1**

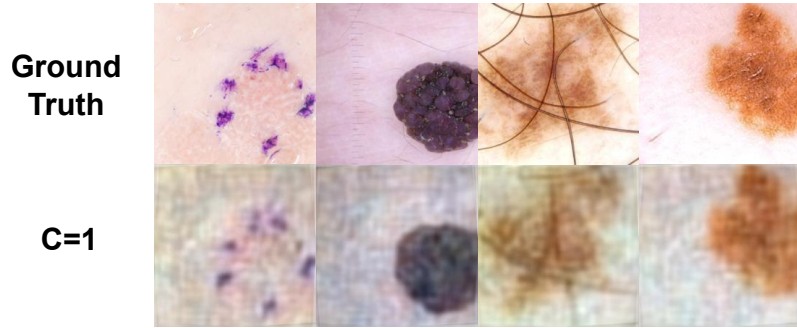

Figure 9: Example reconstructions of SEER trained on ImageNet with $B = 64$ and the Bright property and applied on ISIC2019.

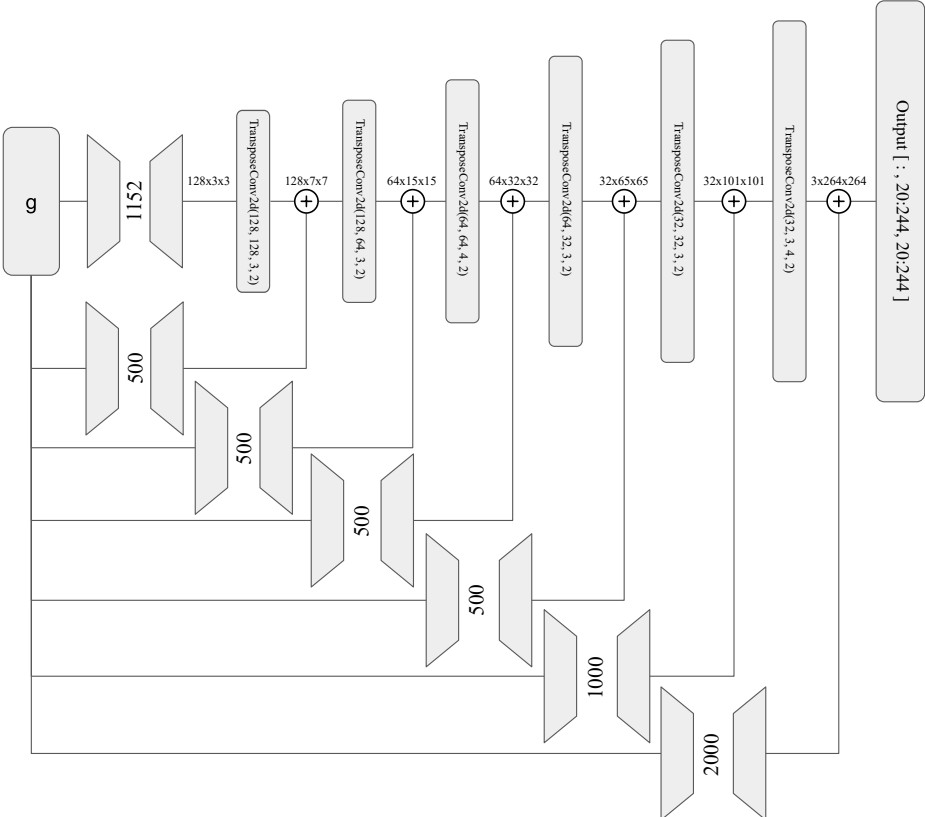

Figure 10: The architecture of our U-Net-based reconstructor $r$ used in our ImageNet experiments. Here $g$ is the randomly subsampled model gradient and the output is the resulting reconstructed image.

In this section, we first describe some implementation details for our ISIC2019 domain shift experiment originally presented in Table 3 that were omitted from the main text for brevity. In particular, to apply our CIFAR10 network trained with the Red property on $B = 128$ to the ISIC2019 dataset, we resized the ISIC2019 images so that their larger side is 350 pixels, then applied a random $224 \times 224$ crop, followed by another downsizing of the image to $32 \times 32$. Since the domain naturally includes a lot of red images, as the data represents pictures of skin conditions, we saw that our method generated too brightly red images. To avoid this issue, we applied the fix presented in App. F.4 for the *Fog* and *Contrast* corruptions. Note that as before, the fix requires no additional knowledge about the color distributions of the images in the ISIC2019 dataset.

Next, we explore a high-resolution version of our domain shift experiment presented in Table 3. In particular, we use the ImagetNet network trained in App. F.10 and apply it to the ISIC2019 dataset prepared like above but with final resolution $224 \times 224$ and only $B = 64$. We successfully recover data from $76.3\%$ of client batches with average PSNR $20.0 \pm 1.8$ and PSNR Top of $21.7 \pm 0.8$. We further show example reconstructions of the images in Fig. 9. Similarly to Table 3, we obtain slightly worse reconstructions compared to the dataset used for the training, but we see that the skin conditions in Fig. 9 are clearly identifiable from our reconstructed images.

## G  U-NET-BASED IMAGE RECONSTRUCTOR

We explain the architecture of our image reconstructor $r$ used in our ImageNet experiments in Sec. 5. Our architecture is inspired by the decoder portion of a U-Net Ronneberger et al. [2015], which has been demonstrated to be a memory-efficient architecture for generating images.

We show our architecture in Fig. 10. In the figure, $g$ depicts our model's gradient subsampled randomly so that only $3\%$ of its entries are kept (See App. I.1). There are two main differences

between our $r$ in Fig. 10 and the original 6-layer U-Net architecture. First, we use no activation functions, thus creating a (sparse) linear reconstructor function $r$. This allows us, similarly to our CIFAR10/100 experiments, to combine $r$ and $d$ into a single linear layer, whose bias becomes the target for our filtered-out inputs $\boldsymbol{X}_{\text{nul}}$. Second, as our architecture does not include U-Net-style encoder, the U-connections of our reconstructor are substituted by pairs of linear layers with a bottleneck in the middle applied on $\boldsymbol{g}$. In Fig. 10, we depict the bottleneck sizes and transposed convolution sizes, as well as the intermediate output sizes of the different layers. The bottlenecks ensure that the memory efficiency of our method is preserved and are inspired by the intuition that the U-connections only need to provide high-frequency content which can live in a much lower-dimensional subspace. Finally, we note that for the purpose of pretraining, we used the first 3 channels of the third transposed convolution layer as our downsized image output and that our transposed convolution stack produces images of size $264 \times 264$, which we then center-crop to produce our ImageNet-sized final output.

## H  HYPERPARAMETERS

In this section, we provide more details about the exact hyperparameters used in our experiments in Sec. 5. We implemented SEER in Pytorch 1.13. Throughout our experiments, we used the Adam optimizer with a learning rate of $0.0001$. For CIFAR10/100, we trained between 500 and 1000 epochs, where an epoch is defined to be 1000 sampled batches from our trainset. To stabilize our training convergence, we adopted gradient accumulation and, thus, updated our modules' parameters only once every 10 gradient steps amounting to 100 gradient descent steps per epoch for those datasets. For ImageNet, we additionally execute a pretraining stage on a downsized version of the training images allowing the network to first learn to recover large details in the image before recovering finer details. For ImageNet, we train with the original $\ell_2$-based loss $\mathcal{L}_{\text{rec}}$ for the first 200 epochs, followed by 300 epochs of using $\ell_1$ version of it, resulting in better visual quality of the reconstruction. At each epoch we only use 4000 randomly sampled batches instead of the full dataset to reduce computational complexity.

For faster convergence and better balance in the optimized objective $\mathcal{L} = \mathcal{L}_{\text{rec}} + \alpha \cdot \mathcal{L}_{\text{nul}}$, we adopted a schedule for the hyperparameter $\alpha$, following an exponential curve of the epoch $\kappa$. The schedule is defined as: $\alpha(\kappa) = min(|B|, 2^{\beta(\kappa)})$, where $\beta(\kappa) = \frac{(K-\kappa)\beta_0 + \kappa\beta_1}{K}$ linearly interpolates between $\beta_0$ and $\beta_1$ across the total number of epochs $K$ with $(\beta_0, \beta_1)$ set to $(-2, log_2|B|)$. For ImageNet, we set $(\beta_0, \beta_1)$ to $(-5, 5.3)$ to allow for better reconstruction earlier in the training process. Finally, we point out that in the multi-client setting, we estimate the cumulative density functions before training for the first and second-highest brightness in a batch on 20000 randomly sampled batches from the trainset (see App. C).

## I  IMPLEMENTATION DETAILS

### I.1  SUBSAMPLED GRADIENTS

In order to save memory and computation, we use only part of the entries in our full model gradient $\boldsymbol{g}$ to construct our intermediate disaggregation space $\mathbb{R}^{n_d}$. In particular, we randomly sample $0.1\%$ of the gradient entries of each of the model's parameters while ensuring that at least 8400 entries per parameter are sampled for our CIFAR10/100 experiments, and $2\%$ and at least 9800 entries for our ResImageNet experiments. This results in $1.6\%$ of the total gradient entries for CIFAR10/100 and $3.0\%$ for ResImageNet. We theorize that we are able to reconstruct nearly perfectly with such a small percent of the gradient entries because there is large redundancy in the information different gradient entries provide.

### I.2  EFFICIENTLY COMPUTING THE DISAGGREGATION LOSS $\mathcal{L}_{\text{NULL}}$

Computing $\mathcal{L}_{\text{null}}$ directly for large batch sizes $B$ takes a lot of memory due to the need to store $\boldsymbol{g}_i$ for all $i$ in the large set $I_{\text{nul}}$. Note that the reason for this is that we want to enforce all of the individual gradient $\boldsymbol{g}_i$ to fall in the null space of $\theta_d$ separately. In practice, to save space, we enforce the same

Table 17: Large batch reconstruction on the bright and red properties from batches of different sizes $B$ on CIFAR10. We report several additional quality of reconstruction metrics—the average MSE and its standard deviation across all reconstructions (*MSE All*), and on the top $37\%$ images (*MSE Top*), as well as, the average LPIPS and its standard deviation across all reconstructions (*LPIPS All*), and on the top $37\%$ images (*LPIPS Top*).

| | CIFAR10, Red | | | | CIFAR10, Bright | | | |
|---|---|---|---|---|---|---|---|---|
| $B$ | MSE Top $\downarrow$ | MSE All $\downarrow$ | LPIPS Top $\downarrow$ | LPIPS All $\downarrow$ | MSE Top $\downarrow$ | MSE All $\downarrow$ | LPIPS Top $\downarrow$ | LPIPS All $\downarrow$ |
| 64 | $0.0009 \pm 0.0002$ | $0.0068 \pm 0.0165$ | $\mathbf{0.039 \pm 0.012}$ | $0.128 \pm 0.194$ | $0.0007 \pm 0.0002$ | $0.0048 \pm 0.0099$ | $\mathbf{0.046 \pm 0.026}$ | $\mathbf{0.104 \pm 0.099}$ |
| 128 | $\mathbf{0.0008 \pm 0.0002}$ | $0.0032 \pm 0.0072$ | $0.050 \pm 0.014$ | $0.096 \pm 0.089$ | $0.0007 \pm 0.0002$ | $\mathbf{0.0031 \pm 0.0060}$ | $0.057 \pm 0.025$ | $\mathbf{0.104 \pm 0.081}$ |
| 256 | $\mathbf{0.0008 \pm 0.0002}$ | $\mathbf{0.0029 \pm 0.0053}$ | $0.052 \pm 0.017$ | $\mathbf{0.095 \pm 0.078}$ | $\mathbf{0.0006 \pm 0.0002}$ | $0.0033 \pm 0.0075$ | $0.054 \pm 0.025$ | $0.109 \pm 0.087$ |
| 512 | $0.0010 \pm 0.0002$ | $0.0035 \pm 0.0051$ | $0.089 \pm 0.025$ | $0.141 \pm 0.081$ | $0.0024 \pm 0.0007$ | $0.0070 \pm 0.0091$ | $0.168 \pm 0.058$ | $0.204 \pm 0.089$ |

condition by computing the surrogate:

$$\widehat{\mathcal{L}}_{\text{nul}} = \| \frac{1}{|I_{\text{nul}}|} \sum_{i \in I_{\text{nul}}} d(\boldsymbol{g}_i) \|_2^2 + \| d(\boldsymbol{g}_{j \sim I_{\text{nul}}}) \|_2^2,$$

where the first part of the equation enforces the mean gradient, and the second part enforces a different randomly chosen gradient at every SGD to both approach 0. This, in practice, has a similar result to the original loss $\mathcal{L}_{\text{null}}$ in that, over time, all gradients in $I_{\text{nul}}$ go to 0.

## I.3 TRAINSET DATA AUGMENTATION

For the purpose of training our encoder-decoder framework, we observed data augmentation of our auxiliary dataset is crucial, especially for large batch sizes $B$. We theorize that the reason for this is the lack of diversity in the reconstruction samples $\boldsymbol{X}_{\text{rec}}$. In particular, as $B$ grows, an increasingly smaller set of images are selected to be the brightest or darkest of any batch sampled from the training set. To this end, when sampling our training batches for CIFAR10/100, we first apply random ColorJitter with brightness, contrast, saturation, and hue parameters 0.2, 0.1, 0.1, and 0.05, respectively, followed by random horizontal and vertical flips, and random rotation at $N * 90 + \epsilon$ degrees, where $N$ is a random integer and $\epsilon$ is chosen uniformly at random on $[-5, 5]$. For ResImageNet, we additionally do a random cropping of the original image to the desired size of $224 \times 224$ before the other augmentations.

## I.4 TRAINSET BATCH AUGMENTATION

As detailed in App. C, when mounting SEER in the secure aggregation setting we select $\tau$ based on a mix of the global and local distributions of $m$. As noted in Sec. 4.1, the probability of attack success with an optimal threshold $\tau$ based on the global distribution of $m$ is $\frac{1}{e}$ in the limit of the number of images being aggregated. We expect for large $B$, therefore, SEER to also successfully reconstruct only for $\approx \frac{1}{e}$ of the securely-aggregated batches, forcing us to avoid training on the rest $1 - \frac{1}{e} > \frac{1}{2}$ of the securely-aggregated batches, which act as a strong noise during training and prevent convergence. This, in turn, results in rejecting training on a big portion of our sampled client batches.

To address this sample inefficiency, we use batch augmentation during training to transform the client batches to ones with a desired brightness distribution. The batch augmentation simply consists of adjusting the brightnesses of individual images within each batch. We do two types of batch augmentations based on two different distributions—one where it contains precisely one image in the batch with brightness above the threshold $\tau$ and another where precisely zero images in the batch have brightness above the threshold $\tau$. We alternate the two augmentations at each step of the training procedure. To achieve the distributions, we adjust all image brightness within a batch using a heuristic method. The method first adjusts the brightness of the most bright image (the least bright image in the case of the dark image property) such that it lands on the desired side of the threshold $\tau$. However, as after adjusting the image, the brightnesses within the batch are no longer normalized, we then need to renormalize the batch, resulting in a new batch brightness distribution. If our new distribution is as desired, we stop. Otherwise, we iterate the process until convergence.

Table 18: Large batch reconstruction on the bright and red properties from batches of different sizes $B$ on CIFAR100. We report several additional reconstruction quality metrics—the average MSE and its standard deviation across all reconstructions (*MSE All*), and on the top $37\%$ images (*MSE Top*), as well as, the average LPIPS and its standard deviation across all reconstructions (*LPIPS All*), and on the top $37\%$ images (*LPIPS Top*).

| | CIFAR100, Red | | | | CIFAR100, Bright | | | |
|---|---|---|---|---|---|---|---|---|
| $B$ | MSE Top $\downarrow$ | MSE All $\downarrow$ | LPIPS Top $\downarrow$ | LPIPS All $\downarrow$ | MSE Top $\downarrow$ | MSE All $\downarrow$ | LPIPS Top $\downarrow$ | LPIPS All $\downarrow$ |
| 64 | $0.0007 \pm 0.0002$ | $0.0021 \pm 0.0046$ | $0.034 \pm 0.010$ | $0.056 \pm 0.059$ | $0.0006 \pm 0.0002$ | $0.0028 \pm 0.0047$ | $0.050 \pm 0.023$ | $0.094 \pm 0.075$ |
| 128 | $0.0007 \pm 0.0001$ | $0.0019 \pm 0.0039$ | $\mathbf{0.031 \pm 0.011}$ | $\mathbf{0.051 \pm 0.061}$ | $0.0010 \pm 0.0003$ | $0.0035 \pm 0.0045$ | $0.112 \pm 0.035$ | $0.164 \pm 0.076$ |
| 256 | $0.0008 \pm 0.0002$ | $0.0022 \pm 0.0047$ | $0.047 \pm 0.014$ | $0.079 \pm 0.064$ | $\mathbf{0.0003 \pm 0.0001}$ | $\mathbf{0.0017 \pm 0.0031}$ | $\mathbf{0.042 \pm 0.019}$ | $\mathbf{0.091 \pm 0.068}$ |
| 512 | $\mathbf{0.0005 \pm 0.0001}$ | $\mathbf{0.0014 \pm 0.0025}$ | $0.034 \pm 0.009$ | $0.057 \pm 0.049$ | $0.0006 \pm 0.0002$ | $0.0032 \pm 0.0043$ | $0.077 \pm 0.032$ | $0.148 \pm 0.085$ |

Table 19: Reconstruction from securely aggregated updates on the bright and dark properties using different numbers of clients $C$ on CIFAR10, for different total numbers of images. We report additional reconstruction quality measures—the average MSE (*MSE Top*) and LPIPS (*LPIPS Top*) and their respective standard deviations on the top $37\%$ images.

| | $C = 4$, Dark | | $C = 4$, Bright | | $C = 8$, Dark | | $C = 8$, Bright | |
|---|---|---|---|---|---|---|---|---|
| #Imgs | MSE Top $\downarrow$ | LPIPS Top $\downarrow$ | MSE Top $\downarrow$ | LPIPS Top $\downarrow$ | MSE Top $\downarrow$ | LPIPS Top $\downarrow$ | MSE Top $\downarrow$ | LPIPS Top $\downarrow$ |
| 64 | $0.0020 \pm 0.0013$ | $\mathbf{0.110 \pm 0.052}$ | $0.0027 \pm 0.0022$ | $\mathbf{0.091 \pm 0.060}$ | $0.0031 \pm 0.0023$ | $\mathbf{0.111 \pm 0.059}$ | $\mathbf{0.0026 \pm 0.0021}$ | $0.123 \pm 0.073$ |
| 128 | $0.0018 \pm 0.0013$ | $0.130 \pm 0.050$ | $0.0028 \pm 0.0018$ | $0.105 \pm 0.064$ | $0.0028 \pm 0.0023$ | $0.118 \pm 0.070$ | $0.0031 \pm 0.0026$ | $\mathbf{0.111 \pm 0.075}$ |
| 256 | $\mathbf{0.0012 \pm 0.0008}$ | $0.130 \pm 0.045$ | $\mathbf{0.0022 \pm 0.0012}$ | $0.113 \pm 0.049$ | $0.0023 \pm 0.0014$ | $0.130 \pm 0.060$ | $0.0035 \pm 0.0022$ | $0.164 \pm 0.085$ |
| 512 | $\mathbf{0.0012 \pm 0.0008}$ | $0.142 \pm 0.054$ | $0.0030 \pm 0.0013$ | $0.208 \pm 0.064$ | $\mathbf{0.0015 \pm 0.0010}$ | $0.159 \pm 0.054$ | $0.0029 \pm 0.0017$ | $0.170 \pm 0.074$ |

## I.5 DETAILS ON PRIOR WORK COMPARISON

In this section, we provide further details about the exact setting in which we do the comparison against prior work in Table 4 and Fig. 1 in the main text. We focus on the feature fishing variant of Wen et al. [2022], that targets batches containing a large percentage of repeated labels, as this setting was shown in prior work [Yin et al., 2021; Geng et al., 2021] to be significantly harder to solve. For the comparison itself, we evaluate both the Fishing and SEER models on client batches of the same class. This setting favors Wen et al. [2022] over SEER, as the Fishing's weights are specifically adapted to it and the SEER model was only trained on batches with randomly selected mixed labels, as in the rest of the paper. We select D-SNR threshold of 5 for the experiment in Table 4 based on Fig. 1. For the comparison against Zhang23 [Zhang et al., 2023] and LOKI [Zhao et al., 2023], we observe they have practically infinite T-SNR, making them trivially detectable.

## J ADDITIONAL MEASUREMENTS OF THE QUALITY OF OUR RECONSTRUCTED IMAGES

In Sec. 5 and App. F.1, we focussed on reporting the quality of our image reconstructions in terms of the popular PSNR image quality metric. In this section, we provide additional image quality measurements in terms of the mean square error of the individual pixels (*MSE*) and the learned perceptual image patch similarity (*LPIPS*) metrics. We present the additional measurements for all large batch experiments from App. F.1 in Table 17 and Table 18 for CIFAR10 and CIFAR100, respectively. Further, in Table 19, we present the additional measurements for the multi-client experiments originally presented in Sec. 5. The additional measurements reinforce our observations from Sec. 5 that SEER consistently reconstructs client data well.

