# OpenReview forum: "Hiding in Plain Sight: Disguising Data Stealing Attacks in Federated Learning"
_ICLR.cc/2024/Conference — ICLR 2024 poster_

### Official Review · Reviewer_ZL2M · 2023-10-21

**Soundness:** 1 poor
**Presentation:** 3 good
**Contribution:** 2 fair
**Rating:** 5
**Confidence:** 4

**Summary:**

This paper points out that malicious attacks for data stealing of FL clients are easily detectable, and propose a novel data stealing attack named SEER that can evade defenses and achieve good data reconstruction performance.

**Strengths:**

Propose to design defense-aware data stealing attacks, making the attacks potentially stealthier.

**Weaknesses:**

1.	Threat model is missing. For example, it is until the description of the SEER attack that the authors mention the availability of auxiliary data at the server, which is not always required by other attacks and can be a strong assumption.
2.	Lack of details: it is not clear what are the exact client-side checks the malicious server needs to evade; it is also not clear what do the authors mean by “handcrafted modifications”. Why is SEER’s mode not handcrafted? The descriptions of the necessary conditions at the end of section 3 are too vague, not to mention providing any theoretical or experimental proofs for those.
3.	It is not clear how SEER exactly breaks through secure aggregation.
4.	Quality and quantity of auxiliary data are key to the performance of SEER. More discussions and ablation studies are needed on auxiliary data.
5.	Complexity issue. Algorithm 1 needs to be repeated for each sample in the batch (i.e., retrain a set of models for each property). This incurs high computational complexity, and should be evaluated carefully and compared with baselines.
6.	No evaluations of SEER and baselines under defenses, which is supposed to be the main motivation of the paper.
7.	The results in Table 3 are quite counter-intuitive and curious. This needs more exploration and explanation.
8.	Only comparison with one baseline in Table 4, which is insufficient. The reviewer would like to see more evaluations and comparisons with the malicious server attacks mentioned in the "gradient leakage attacks" subsection in introduction, and "malicious server attacks" sub-section in related work. Specifically, comparisons with the following recent attacks are needed:

1) Joshua C. Zhao, Atul Sharma, Ahmed Roushdy Elkordy, Yahya H. Ezzeldin, Salman Avestimehr, and Saurabh Bagchi. Secure aggregation in federated learning is not private: Leaking user data at large scale through model modification. arXiv, 2023. (Just accepted to S&P 2024 with the title "LOKI: Large-scale Data Reconstruction Attack against Federated Learning through Model Manipulation").

2) Shuaishuai Zhang, Jie Huang, Zeping Zhang, and Chunyang Qi. Compromise privacy in large-batch federated learning via malicious model parameters. In ICA3PP, 2023.

**Questions:**

1.	Does a distinguishing property for each sample in a batch always exist? What if it does not exist? How does it affect the performance of SEER?
2.	Are all the experiments under any kinds of defenses or nor? How do the experiment results show that SEER is undetectable?

---

> ### Author Response · Authors · 2023-11-19
> **Response to Reviewer ZL2M (1/2)**
>
> $\newcommand{Rfive}{\textcolor{purple}{ZL2M}}$
> We thank the reviewer $\Rfive$ for their extensive feedback; our responses are below. We will incorporate the clarifications and explanations below in future revisions of our paper. We are happy to continue the discussion in case more questions arise.
>  **GQ# refers to points in our general response.**
>
> **Q5.1: Can you include the auxiliary data requirement in the threat model and provide proofs?**
>
> We acknowledge that this requirement could be already mentioned in Sec. 3 and will include this in the next revision. Our ablations with aux. data (see **Q5.4**) show that this is not restrictive. This contrasts with other aspects mentioned in Sec. 3, such as architecture modifications, which are trivially detectable yet still considered viable by many prior works. We believe our threat model is much more restrictive for the attacker and more realistic, and the success of SEER despite this hardening, is our key contribution.
>
> We are not sure what the reviewer refers to as proofs for the threat model—if the reviewer can elaborate, we are happy to discuss further.
>
> **Q5.2: What client-side checks does the attacker need to evade, and how do you show that SEER is undetectable?**
>
> We argue in the paper that the attacker should evade all checks known to practitioners at that point in time (a standard security setup). This was ignored in prior work, inflating the perceived danger from those attacks, which we show are easily detectable. In our paper, SEER is benchmarked and is not detected by D-SNR (Figure 1, Table 4) and the transmission weight-space detection (App. A)---two principled checks that we show can detect all prior work in our threat model. Importantly (see Outlook), we do not claim that no future detection method will be able to detect SEER, and we encourage work on complex yet principled detection methods.
>
> **Q5.3: What is the meaning of “handcrafted modifications” and why is SEER not in this group?**
>
> As “handcrafted modifications” we dub manual changes of the model, such as setting weights to 0 or 1 “by hand”, as these often directly imply trivial detectability—we explain the reasons in the paper. SEER only changes weights via continuous optimization, thus has no handcrafted changes—while this makes SEER the only attack not detectable at the moment, as we mention in **Q5.2** and the paper, it is possible that future work changes this, but we see no easy path there. We do agree that “handcrafted modifications” is not a rigorous mathematical definition, but we still find it useful as a means to accurately discuss prior work.
>
> **Q5.4: Can you clarify how SEER breaks secure aggregation?**
>
> On a high level, breaking secure aggregation reduces to disaggregating individual gradients from their aggregated sum/mean and is thus not fundamentally different from attacks on large batches when no batchnorm (BN) layers are present in the network (See **Q2.3** for what happens under BN). Disaggregation of individual sample gradients from their sum is possible in the MS setting in practice, as was shown in prior work, e.g., [Fowl et al., 2022b]. Our contribution is an attack that is not easily detectable in gradient and weight space (See Sec. 3). We provide further technical details on the variant of SEER we use for secure aggregation in App. B.
>
> **Q5.5: Can you provide more ablation studies on the size and type of auxiliary data?**
>
> Beyond Table 3, prompted by this question we show results of two new experiments: in **GQ4** we investigate a strong distribution shift where clients use medical data, and in **GQ5** we ablate the size of the aux. corpus. In both cases, we find that the aux. data assumption of SEER is not restrictive.
>
> **Q5.6: Is it necessary to train a separate SEER model for many properties?**
>
> No. A SEER model trained once can be applied across different clients, training rounds, and even FL deployments and will, whenever successful (88-98% of the time in the settings of our experiments), produce high-quality reconstructions of private user images that satisfy the property. We argue, in sync with prior work, that privacy is an absolute property—as long as any private client image is leaked to attackers, this is a privacy leak and breaks the key promise of FL. A good analogy is security, where a similar perspective is widely accepted.
>
> **Q5.7: Can you evaluate SEER under defenses?**
>
> Certainly, we agree that this is important. See **GQ3**, where we, prompted by this question, perform a new experiment and show that SEER still succeeds under differential privacy. Additionally, we point the author to Figure 1, Figure 4 and Appendix A, where we evaluate SEER under two principled client-side checks that we developed and confirm they are very effective as defenses against most prior work—yet, they do not break SEER.

---

> ### Author Response · Authors · 2023-11-19
> **Response to Reviewer ZL2M (2/2)**
>
> $\newcommand{Rfive}{\textcolor{purple}{ZL2M}}$
> **Q5.8: Table 3 is counterintuitive—can you provide an explanation and more exploration?**
>
> As more exploration we offer new experiments on distribution shifts with medical data (**GQ4**) which support our prior conclusions. We have failed to understand what in Table 3 the reviewer found counterintuitive, and would gladly discuss further if the reviewer could elaborate.
>
> **Q5.9: Does the property always exist for each sample in the batch; how does this affect SEER?**
>
> Interesting question. Rigorously speaking, yes—for each image $x$, the property “equal to $x$” will single out only this image. So the actually interesting question is can each image be singled out by a property that is (i) simple enough to well train malicious weights (ii) transferable to other batches without retraining, as this is one of the key features of SEER (see **Q4.6**). As we do not target particular images but simply aim to demonstrate that _some_ of the user’s private images can be leaked (a standard understanding of privacy; see **Q5.6**), this question has no immediate impact on SEER. Whatever property we chose in our experiments, a once-trained SEER model was able to recover images from many batches with high success (App. C.6).
>
> **Q5.10: Can you compare with more MS attacks such as Zhang23 and Zhao23 (LOKI)?**
>
> As we note in the paper, these and similar works either focus on much stronger threat models or are easily detectable, thus we believe it is not appropriate to include them in Sec. 5. Regardless, prompted by this question, we ran two new experiments following the setup of Table 4, obtaining expected results.
>
> Recall that in Table 4 SEER obtains UndRec% of 90 (in 90% the attack is undetectable, as we always have DSNR<5 and the transmission metric<0.3) and also obtains good reconstructions.
>
> **Zhang23**: As noted in Sec. 3, this is a boosted analytical attack, and thus has the following two drawbacks:
> 1. Their transmission (passing inputs unchanged to the first FC layer; necessary for convnets) does not scale to realistic networks such as ResNet due to stride>1 and pooling—accordingly, these results are not present in the cited work. As we cite in our paper, Fowl et al., 2022b show that transmitting this way up to the final pooling already severely downscales the image. We further find that past that pooling (thus pre-first FC), the images can not be transmitted with any success.
>  2. Ignoring the scalability issue, the transmission used in this work is easily weight-space detectable on _any_ convnet (see App. A).
>
> Despite these, we ran Zhang23 in the setting of Table 4, taking max PSNR of batch reconstructions. The results follow:
>
> _Rec%_: **5** | _UndRec%_: **0** | _PSNR_: **15.8 +- 1.8** | _PSNR-und_: **N/A** | _PSNR-und-rec_: **N/A**
>
> As expected, while inputs to the first FC layer were recovered well (average PSNR of 43), this almost never leads to an image reconstruction, i.e., Rec% is only 5. Moreover, the transmission metric (App. A) is $\infty$ in all cases (as opposed to <0.3 for natural networks and SEER), and thus, UndRec% is 0, and the PSNR of undetected reconstructions is undefined.
>
> **Zhao23 (LOKI)** Similarly, as we note in the paper, LOKI focuses on a different (significantly stronger) threat model where architectural changes and sending different updates to clients (_model inconsistency_) are allowed. LOKI focuses on scaling attacks in this setting and succeeds in this goal. However it can, by design, never be directly applied to fixed/known architectures such as ResNet or in cases where clients get the same model.
>
> To recall our Sec. 3, our main focus are practical cases with threat-aware clients, so we require use of known architectures (as clients can easily detect modules that are never used in practice, such as LOKI’s 1Conv+2FC at the start of the network) and no model inconsistency (as this is easily detectable and revertable with reverse secure aggregation [Pasquini et al., 2022]).
>
> Due to the stronger threat model, LOKI can avoid the drawback (1) of Zhang23 stated above, but the issue (2) still stands as LOKI is fundamentally a boosted analytical attack, thus it can be detected. We confirm this experimentally on an augmented ResNet with 1Conv+2FC in front (as LOKI can not be applied in the exact setting of Table 4):
>
> _Rec%_: **100** | _UndRec%_: **0** | _PSNR_: **143.4 +- 10.3** | _PSNR-und_: **N/A** | _PSNR-und-rec_: **N/A**
>
> As expected, adding an FC layer to the front allows almost exact reconstruction, a known fact in the field [Phong et al., 2018; Geiping et al., 2020]. However, transmission is still needed, thus UndRec%=0, i.e. no images are reconstructed undetectably. Additionally, any threat-aware client could notice FC layers in front of the convolutional stack, which never occurs naturally.
>
> We hope these results help the reviewer acknowledge our points about these attacks.

---

> > ### Comment · Reviewer_ZL2M · 2023-11-22
> >
> > Thanks for the response, which addresses some of my concerns. The reason why a distinct property exists for each sample exist is still not clear to me.

---

> ### Author Response · Authors · 2023-11-22
> **Follow-up to Q5.9**
>
> $\newcommand{Rfive}{\textcolor{purple}{ZL2M}}$
> We thank the reviewer $\Rfive$ for engaging with the rebuttal and following up on Q5.9. We elaborate on our answer above, and explore some related questions:
>
> **(1) For a given d-dim example $x_i$ in a batch of size B, is there always a property that separates it from the rest of the batch?**
> Above we gave a positive answer in the general case. Here, we note that the answer is still positive even for restricted properties $m(x) < \tau$, where $m$ is a linear measurement, which we introduce on top of Sec 4.1 and use in SEER. The underlying question here is if the hyperplane $m(x)-\tau=0$ can always separate the point $x_i$ from the rest. This is answered positively by a well-known result from theoretical ML (see e.g., [1]) which states that the VC dimension of affine classifiers (in this case $sgn(m(x)-\tau)$) in $\mathbb{R}^d$ is $d+1$. Thus, such a property always exists when our images are in a general position (which usually holds) and $B \leq d+1$ (which holds even for low-dim CIFAR10, as $d+1=3073$ which is far above practical batch sizes).
>
> **(2) Can such a property be efficiently constructed?**
> Yes. This is equivalent to finding _any_ linear hard-margin SVM on linearly separable data, while ignoring the constraint of margin maximization—this can be encoded and exactly solved as a simple linear program (LP) in polynomial time of the input size $d$.
>
> **(3) How does this look like in practice with SEER?**
> In practice, we obviously do not know the images we are trying to reconstruct and do not target any particular image but demonstrate unexpected privacy leakage in FL by showing that some client data can be always leaked in practical deployments. The question of which properties are easier/harder to train SEER with is interesting and we offer an initial exploration in App. C.6 where we test many properties (Bright/Dark, Red/Green/Blue, Hor/Ver Edges, combinations of previous, random properties) and show that SEER can be trained well with all of them to reconstruct substantially different types of images (see Fig. 5).
>
> We hope this clarifies Q5.9. We encourage the reviewer to follow up on any of Q5.1-Q5.10 if there are still outstanding questions, and reassess our paper in light of our answers.
>
> [1] https://mlweb.loria.fr/book/en/VCdimhyperplane.html

---

### Official Review · Reviewer_sMvX · 2023-10-24

**Soundness:** 3 good
**Presentation:** 3 good
**Contribution:** 4 excellent
**Rating:** 8
**Confidence:** 4

**Summary:**

This paper proposes the disaggregation signal-to-noise ratio (D-SNR) metric, designed to detect the vulnerability of local clients to malicious attacks (MS). Subsequently, the authors outline a set of requirements for future realistic MS attacks. They then present a novel attack framework, SEER, which operates by disaggregating the gradients in a hidden space. Specifically, the SEER framework encompasses a shared model dispersed among clients for encoding gradients, a server-side disaggregator to nullify the contributions of images not satisfying the pre-defined property and a server-side reconstructor to reconstruct images that comply with the said property. The framework facilitates end-to-end training and presents a challenge for detection using D-SNR, in contrast to previous MS methods. Experiments also show the effectiveness of the proposed method.

The main contributions of this work include a pioneering study of client-side detectability, the introduction of a novel detection metric, highlighting potential concerns for future MS, and the unveiling of a groundbreaking realistic MS approach.

**Strengths:**

- The paper is well-written and easy to follow, making for an enjoyable read.
- The motivation is clear and the study problem is significant.
- The proposed detection metric and attack framework are innovative and intriguing, offering a fresh viewpoint on the privacy issue in FL.

**Weaknesses:**

The main concern is about the auxiliary data used to train the SEER framework. The key point is that the pre-defined property could separate examples.
The paper details experiments where CIFAR10 serves as the auxiliary data for other client datasets (CIFAR10.1v6, CIFAR10.2, and TinyImageNet). While the results affirm its robustness to distributional disparities between the auxiliary and client data, there may be scenarios where this approach is less effective. For instance, when the clients possess specialized data such as medical records, the server have no prior knowledge about the data and use generic dataset like CIFAR10 as the auxiliary data, the framework might encounter challenges, because the pre-defined property might significantly differ across such diverse datasets.

**Questions:**

- How does the proposed SEER MS fare when differential privacy mechanisms are implemented?
- How does SEER perform in non-iid settings?

**Details Of Ethics Concerns:**

Authors have already discussed this in the paper.

---

> ### Author Response · Authors · 2023-11-19
> **Response to Reviewer sMvX**
>
> $\newcommand{Rfour}{\textcolor{green}{sMvX}}$
> We are honored that the reviewer $\Rfour$ found both our proposed attack and defense techniques innovative and intriguing. We appreciate the raised questions which prompted us to conduct additional experiments, shown in our general answer. Below we provide pointers to those experiments. We are happy to continue the discussion in case of additional questions. **Note that GQ# refers to points in our general response.**
>
> **Q4.1: Can you provide additional distribution shift experiments where clients have specialized (e.g., medical) data, and SEER is trained on a generic dataset (CIFAR10)?**
>
> In **GQ4**, we report the results of exactly the experiment described by the reviewer on the ISIC2019 dataset. As on this dataset, our method performs even better than on TinyImageNet, we remain confident that SEER is robust regarding the nature of the client data.
>
> **Q4.2: Can differential privacy defend against SEER?**
>
> We agree that this is an important aspect to evaluate. See our new experiment in **GQ3** prompted by this question, where we demonstrate that DP is unfortunately not an effective defense.
>
> **Q4.3: How does SEER perform in non-IID settings?**
>
> See our new experiment in **GQ2** prompted by this question, where we notice no significant degradation in the non-IID case.

---

> > ### Comment · Reviewer_sMvX · 2023-11-21
> >
> > Thanks for your reply. Currently, I have no questions, will keep my rating, and will consider other reviewers' discussions.

---

### Official Review · Reviewer_SRu7 · 2023-10-29

**Soundness:** 3 good
**Presentation:** 3 good
**Contribution:** 2 fair
**Rating:** 6
**Confidence:** 2

**Summary:**

1. The paper focuses on the issue of client-side detectability of malicious server (MS) attacks in federated learning. It discusses prior work on gradient leakage attacks in federated learning, including honest server attacks, malicious server attacks, and the limitations of existing attacks.

2. To ensure reproducibility, the authors provide the source code of SEER in the supplementary material and detail how to install the code prerequisites and reproduce the experiments presented in the paper.

3. Overall, the paper contributes a novel attack strategy that avoids detection in federated learning, highlights the importance of attack detectability, provides insights into reconstructing data from securely aggregated gradient updates, and addresses the ethical implications of their attack.

**Strengths:**

1. The authors propose a novel attack strategy called SEER that avoids detection while effectively stealing data despite aggregation. They demonstrate that all prior attacks are detectable in a principled way and highlight the importance of studying attack detectability.

2. SEER is designed to reconstruct data from securely aggregated gradient updates. The authors describe in detail how they combine local and global distribution information about the client data to achieve this reconstruction.

3. The paper also discusses the ethical implications of their attack and acknowledges the potential disparate impact and privacy risks. However, they argue that their investigation of detection and emphasis on realistic scenarios have a positive impact by enabling further studies of defenses and helping practitioners better estimate privacy risks.

**Weaknesses:**

1. Would the author have some initial idea for defensing the proposed attacks？

**Questions:**

See weakness part

---

> ### Author Response · Authors · 2023-11-19
> **Response to Reviewer SRu7**
>
> $\newcommand{Rthree}{\textcolor{red}{SRu7}}$
> We thank the reviewer $\Rthree$ for their overall positive view of our work. We are pleased that they found our defense strategies principled and important and appreciated our discussion of the impact of our work on privacy and fairness of FL. We answer outstanding questions below. We are happy to continue the discussion in case of additional questions. **Note that GQ# refers to points in our general response.**
>
> **Q3.1: Can you outline possible mitigation strategies?**
>
> Please see our new experiment in **GQ3**, where we show that commonly used differential privacy is not an effective defense against SEER. Further, as we point out in the paper, we find the restriction of protocols to disallow obvious malicious modifications, such as changing the architecture or sending different updates to clients, to be the most important precaution to take against privacy attacks in the MS setting. Beyond this, we find principled client-side detection to still be the most promising direction (as also discussed in **Q2.2**). While none of the currently existing defenses work against SEER, we hope that our paper further emphasizes the need for research on detection and are looking forward to new results in this area.

---

> > ### Comment · Reviewer_SRu7 · 2023-11-22
> > **Response to Authors**
> >
> > Thanks for rebuttal. I have no more concerns and will maintain my score to support acceptance.

---

### Official Review · Reviewer_dXt6 · 2023-10-30

**Soundness:** 2 fair
**Presentation:** 3 good
**Contribution:** 3 good
**Rating:** 6
**Confidence:** 3

**Summary:**

This paper studies data-stealing attacks in federated learning with malicious servers. The authors first demonstrate the client-side detectability of existing attacks by introducing a simple vulnerability metric called D-SNR, and then reveal the limitation of their attack design which is their dependence on the underlying honest attack. Based on this analysis, the authors further propose a new attack framework SEER using a server-side decoder that is jointly optimized with the shared model to reduce the chances of being detected. Experiments on three image datasets show that the proposed attack can successfully extract user data of large batch sizes (up to 512) even under secure aggregation.

**Strengths:**

- The paper is in general well-written and self-contained and provides a good summarization of prior studies in this field.

- The proposed method addresses several limitations of existing attacks including client-side detectability and the assumptions on BN statistics and labels.

- The idea of jointly learning a disaggregator and a reconstructor with the shared model is novel.

- The empirical results are in favor of the proposed method in terms of stealthiness and reconstruction quality.

**Weaknesses:**

- The work lacks a more nuanced discussion on the threat model. Despite having relaxed some of the assumptions of previous work, the proposed attack is still somewhat restrictive in the sense that it requires multiple steps of offline training and an auxiliary dataset of sufficient size, and it is not very clear how to deploy such an attack during the federated learning process in practice. In particular, the joint optimization of the shared model seems to ignore the classification loss of the federated task, which might be leveraged to detect the attack in a retrospective way (e.g., a client may evaluate the local training loss before and after each update and choose to opt out of training if the loss is not reduced after several rounds). One reasonable strategy is to deploy the attack in the first round of training, i.e., substitute the random initialization with the optimized model weights, but this may still raise the alarm if it’s beyond normal weight distributions. It would be better to add corresponding discussions and limitations.

- It is not quite clear how to utilize the batch normalization statistics to design the local selection strategy of the property.

- The proposed attack algorithm appears to be quite computationally expensive (~14 GPU days to train for ResImageNet). It also implicitly assumes knowledge of the private data distribution as is for all learning-based attack approaches that rely on a large auxiliary dataset.

**Questions:**

1. Could you provide some further explanation on how SEER utilizes batch normalization to choose the local property? In cases where this is not feasible (e.g., secure aggregation or no batch norm layer), would it be possible for the attacker to simply run a linear search to find the optimal property for each batch?

2. Would it be possible to replace the learned disaggregating mapping with some hand-crafted criteria such as simply recovering the image with the largest loss within the batch?

3. Does the design of the local properties take into account the potential non-IIDness of the data? What’s the success rate if the clients’ data are locally correlated?

4. Could you share some insights on designing potential defense and mitigation strategies for the proposed attack besides standard options like DP and HE?

---

> ### Author Response · Authors · 2023-11-19
> **Response to Reviewer dXt6**
>
> We are delighted the reviewer $\textcolor{blue}{dXt6}$  identified our method as addressing many shortcomings in prior state-of-the-art methods and considers our methodology novel. We are thankful for the reviewer’s insightful questions. We provide exhaustive clarifications on SEER’s technical details, as well as pointers to the additional experiments conducted for the reviewer below. We are happy to continue the discussion in case of follow-ups. **GQ# refers to our general response.**
>
> **Q2.1: Are the need for offline training and auxiliary dataset limitations of SEER?**
>
> We believe not. To confirm this, we show that SEER is robust to distribution shifts between client and aux. data (Table 3 and a new experiment in **GQ4**), and that it does not require a large aux. corpus (a new experiment in **GQ5**). Offline training is not a limitation as it is independent of the FL deployment (as we show knowledge of the private data distribution is not needed) and can be thus done at any time before the main FL training starts.
>
> **Q2.2: Can SEER be detected by observing the loss over rounds or comparing its weights to benign model weights?**
>
> SEER does not require application in more than one round, and such attacks are often applied in the first round [Balunovic et al., 2022b], as the reviewer suggests. Further, it is known that in FL, per-client loss values can be noisy, even if the global loss consistently falls. As noted in [Boenisch et al., 2021], this makes loss-based detection unlikely to work. Regarding weights, as SEER only modifies them through continuous optimization, it uses no obvious handcrafted patterns that can be flagged (as opposed to prior work analyzed in the paper). However, as we note in our Outlook section, we do not claim that future refinements of this idea will fail to mitigate SEER, and find this a good direction for future work. As SEER is resistant to all current defenses (see also **GQ3** for our experiment on DP), we believe our work is an important contribution and can motivate future defenses.
>
> **Q2.3: How are BN statistics used to pick the local property? When this is impossible, can we use linear search?**
>
> Unlike prior work, we do not rely on BN statistics, but show that the mere presence of BN harms privacy. Consider the no-BN case, where the final gradient is a mean of sample gradients, and the forward passes (and thus the gradients) of samples are independent. When BN is used, the final gradient is still a simple mean, but the computation of sample gradients is intertwined, i.e., $g(x_i)$ and $g(x_j)$ depend on each other. This is an issue because it enables a MS attacker to influence $g$ to make e.g., the gradient of $x_i$ non-negligible iff $x_i$ is the brightest image in the batch, disaggregating this example. More formally, BN breaks the implicit independence assumption in the calculation of $p(X=1)$ in the proof of Proposition 3.2 in [Wen et al., 2022], allowing to improve on its attack success upper bound. SEER does exactly this, achieving empirically much better success rates than bound’s $\frac{1}{e}$ by training weights to be attackable, incentivizing disaggregation through our loss.
>
> This is impossible without BN as the computational graph of $x_i$ is independent of other samples, thus there is no way to find the “brightest image”. Finding the optimal threshold of the property that the reviewer asks about is independent of this—even with the optimal property, if there is no BN, we cannot get attack success above the bound from [Wen et al., 2022], unlike SEER in the BN case.
>
> **Q2.4: Can we replace the learned disaggregation with handcrafted criteria?**
>
> We believe this is not possible, as designing an undetectable disaggregation scheme is inherently hard. In particular, we distinguish between (1) choosing a property metric $m$ (e.g., Most Bright), which in SEER we do manually (see App. C.6 where SEER is shown to be robust to property choice) and where the sample loss could be used as a criterion, and (2) finding malicious weights that isolate the image with that property. Point (2) is very hard without trivially detectable changes, which is the very motivation for SEER, and what we see as our main contribution.
>
> **Q2.5: How does SEER perform in the case of non-IID client data?**
>
> We do not notice significant degradation in this case—see our new experiments in **GQ2**.
>
> **Q2.6: Can you outline possible mitigation strategies besides DP/HE?**
>
> As we point out in the paper, we find the restriction of protocols to disallow obvious modifications, such as changing the architecture or sending different updates to clients, to be the most important precaution against privacy attacks. Beyond this, we find principled client-side checks to be the most promising direction (as discussed in **Q2.2**, Outlook section). While no existing defenses work against SEER, we hope that our paper emphasizes the need for research on detection and are looking forward to new results in this area

---

> > ### Comment · Reviewer_dXt6 · 2023-11-21
> > **Thanks**
> >
> > Thanks for your response. The reviewer does not have further questions for the moment and will stay with their initial evaluation in support of acceptance.

---

### Official Review · Reviewer_Lg7w · 2023-11-01

**Soundness:** 3 good
**Presentation:** 3 good
**Contribution:** 3 good
**Rating:** 6
**Confidence:** 3

**Summary:**

The paper found that existing malicious server attacks in FL are detectable on the client side using a metric, D-SNR. The authors then propose a new attack called SEER which can bypass the detection by co-optimizing the disaggregator and reconstructor.

**Strengths:**

1. D-SNR is a useful metric to detect disaggregation by malicious server
2. SEER is an attack that can efficiently extract information and bypass D-SNR detection.

**Weaknesses:**

1. The performance of SEER will depend on the task and dataset. For some dataset, SEER might lose effect since it's not always possible to disaggregate a value from the mean.
2. The evaluation is on CIFAR10/100. It'd better to demonstrate the effectiveness of SEER on more tasks closer to real-world applications.

**Questions:**

N/A

---

> ### Author Response · Authors · 2023-11-19
> **Response to Reviewer Lg7w**
>
> $\newcommand{Rone}{\textcolor{orange}{Lg7w}}$
> We are glad the reviewer $\Rone$ found our proposed metric D-SNR useful for detecting existing MS attacks and found our newly proposed attack SEER effective. We address the raised concerns below. We are happy to continue the discussion in case of additional questions. **Note that GQ# refers to points in our general response.**
>
> **Q1.1: Does the performance of SEER depend on the setting?**
>
> Certainly—some settings can be inherently harder for the attacker, e.g., in our Table 1 SEER’s accuracy is always a few % lower in the CIFAR10+Bright than in the CIFAR100+Red setting. However, in both of these settings as well as in our experiments on ResImageNet and ImageNet (see **GQ1**), and across various properties and batch sizes, SEER consistently succeeds in stealing user data. Thus, despite our considerable efforts in making the evaluation thorough, we are yet unable to find a setting where SEER underperforms.
>
> **Q1.2: When and why is it possible to disaggregate individual sample gradients from their mean in the MS setting?**
>
> The typical intuition about the disaggregation of a single element from a sum or a mean (in our case of gradients) is that it is not possible. In federated learning, this is formalized in the notion of Secure Aggregation, which provably secures the privacy of individual sample gradients in the honest-but-curious server setting. However, Secure Aggregation can be attacked in the MS server setting. This is due to breaking a key assumption in the theory that the attacker has no influence over the gradient function $g(x)$. In the MS setting,  $g(x)$ can be influenced partially by choosing the global model weights $\theta_f$ sent to the clients. Prior work has leveraged this observation in practice (e.g. See [Wen et al., 2022]) to disaggregate gradients from their mean almost independently of the client data with very good results. Unfortunately, such work, as discussed in Section 3, is trivially client-detectable. We see the results of SEER across several different properties (App. C.6), and datasets (Section 5) to be a strong indication that the same strong disaggregation results are also possible when the attack is not easily detectable.
>
> **Q1.3: Can you show results on more datasets beyond CIFAR10/CIFAR100?**
>
> We refer the reviewer to our results on ResImageNet that were included in our original submission as a way to investigate high-resolution settings (top of page 8), as well as our new results on ImageNet that were prompted by this reviewer’s question (**GQ1**), and our new distribution shift results on domain-specific medical data (**GQ4**).

---

### Author Response · Authors · 2023-11-19
**General Response**

$\newcommand{Rone}{\textcolor{orange}{Lg7w}}$
$\newcommand{Rtwo}{\textcolor{blue}{dXt6}}$
$\newcommand{Rthree}{\textcolor{red}{SRu7}}$
$\newcommand{Rfour}{\textcolor{green}{gB2C}}$
$\newcommand{Rfive}{\textcolor{purple}{3m7j}}$
We thank all reviewers for their feedback and are pleased that they all share our vision that analyzing and designing MS attacks from the point of view of detectability is important. We are also thrilled that almost all reviewers view our work favourably, acknowledging our main contributions. In the following, we respond to global questions raised by several reviewers (marked **GQ1**-**GQ5**) and present five new experiments that address their concerns, showing that our attack is effective in various challenging settings. These are also included in a new paper revision colored in blue; Appendices C.8 to C.12 and expanded Table 3. In individual responses to each reviewer, we point to our global answers and directly address their remaining questions.

**GQ1: Can you confirm the effectiveness of SEER in more high-resolution settings?**
Yes. We have added a new experiment on the original Imagenet dataset, see App. C.11. We can see that SEER can also recover data well in this setting—our average PSNR is 21.9, and the success rate is above 80%. A visual inspection of the results confirms that the recovered images look as good as our CIFAR10 reconstructions.

**GQ2: Is SEER’s reconstruction quality lower in non-iid settings?**
Not significantly—the attack succeeds even in such cases. Prompted by reviewers’ concerns, we performed a new experiment, shown in Appendix C.10 in the latest revision, where our CIFAR10-trained model was tested on clients with different levels of data heterogeneity, simulated using a Dirichlet distribution over their class distribution. We see that even for highly heterogeneous settings, our method still has a success rate of over 90%.

**GQ3: Can differential privacy defend against SEER?**
SEER is very robust to differential privacy defenses.  In particular, prompted by reviewers’ concerns, we performed an experiment shown in App. C.8 in the latest revision, where we attack gradients defended by DP-SGD with different clipping norms $\mathcal{C}$ and level of noise $\sigma$. We see that our method succeeded non-trivial percent of the time even under severe clipping (up to $\mathcal{C}=1$) and severe noise (up to $\sigma=0.007$), extreme parameter choices which would never be deployed in practice as they would result in significant loss of accuracy for the final model [1].

**GQ4: Can you experiment with more severe distribution shifts e.g., a case where SEER is trained on CIFAR10/ImageNet, but the clients hold medical data?**
Certainly; in App. C.12 in our new revision we provide the results of applying our ImageNet model from **GQ1**, and our CIFAR10 model from Table 3, to the mole classification task from ISIC2019 [Tschandl et al., 2018]. For the ImageNet model, we observe that while the results have as expected degraded slightly due to the severe data shift compared to the original ImageNet experiment, our method still succeeds in reconstructing the data >75% of the time, and individual moles are clearly identifiable. For our CIFAR10 model, we see that our results on ISIC2019 are even better than our results on TinyImageNet, reaffirming the overall robustness of SEER to severe data shifts between clients and auxiliary data.

**GQ5: Does SEER require a large auxiliary dataset to successfully steal client data?**
No, SEER does not require a lot of auxiliary data, as we demonstrate in a new experiment prompted by this question (App. C.9 in the latest revision of our paper). There, we show that models trained only on 5% of CIFAR10’s training set, corresponding to just 2500 images, still allow us to severely compromise the client’s privacy (average reconstruction PSNR >21) more than 80% of the time.

[1] Wei, Wenqi, et al. "A framework for evaluating gradient leakage attacks in federated learning." arXiv preprint arXiv:2004.10397 (2020).

---

### Author Response · Authors · 2023-11-23
**Discussion Phase Ending Soon**

$\newcommand{Rone}{\textcolor{orange}{Lg7w}}$
$\newcommand{Rtwo}{\textcolor{blue}{dXt6}}$
$\newcommand{Rthree}{\textcolor{red}{SRu7}}$ 	$\newcommand{Rfour}{\textcolor{green}{sMvX}}$
$\newcommand{Rfive}{\textcolor{purple}{ZL2M}}$ We thank all reviewers once again for their feedback. We appreciate that most reviewers have acknowledged the rebuttal, and that reviewers $\Rtwo$, $\Rthree$ and $\Rfour$ have explicitly confirmed their support of acceptance. As the discussion phase ends in a few hours, we encourage the remaining reviewers to let us know if they have any outstanding concerns which we are still happy to answer.

---

### Meta-Review · Area_Chair_eC33 · 2023-12-05

**Metareview:**

This submission, titled "Hiding in Plain Sight: Disguising Data Stealing Attacks in Federated Learning", investigates the question of data leakage attacks in federated learning in the malicious-server setting. A question unsolved in previous work was whether these attacks could be launched even against a defender that is aware of malicious-server attacks, i.e. whether the attacks could be hidden. The submission defines criteria for dectability, shows that existing attacks can indeed be reliably detected and then proposes a novel attack that cannot be detected.

**Justification For Why Not Higher Score:**

The attack comes with a few caveats and limitations brought up during the review process, which remain as open questions for future work. It would also be helpful if the authors could include their explanations brought up during the review question answering why separating planes exist that allow for the leakage of individual data points, into the submission.

**Justification For Why Not Lower Score:**

The proposed attack shows that malicious server attacks are practical even against a defender that is aware of the problem of malicious-server attacks.

---

### Decision · Program_Chairs · 2024-01-16

Accept (poster)